# Clock gene-dependent glutamate dynamics in the bean bug brain regulate photoperiodic reproduction

**Masaharu Hasebe** *, **Sakiko Shiga**

Department of Biological Sciences, Graduate School of Science, Osaka University, Osaka, Japan

* h.masaharu@bio.sci.osaka-u.ac.jp

## Abstract

Animals adequately modulate their physiological status and behavior according to the season. Many animals sense photoperiod for seasonal adaptation, and the circadian clock is suggested to play an essential role in photoperiodic time measurement. However, circadian clock-driven neural signals in the brain that convey photoperiodic information remain unclear. Here, we focused on brain extracellular dynamics of a classical neurotransmitter glutamate, which is widely used for brain neurotransmission, and analyzed its involvement in photoperiodic responses using the bean bug *Riptortus pedestris* that shows clear photoperiodism in reproduction. Extracellular glutamate levels in the whole brain were significantly higher under short-day conditions, which cause a reproductive diapause, than those under long-day conditions. The photoperiodic change in glutamate levels was clearly abolished by knockdown of the clock gene *period*. We also demonstrated that genetic modulation of glutamate dynamics by knockdown of glutamate-metabolizing enzyme genes, *glutamate oxaloacetate transaminase* (*got*) and *glutamine synthetase* (*gs*), attenuated photoperiodic responses in reproduction. Further, we investigated glutamate-mediated photoperiodic modulations at a cellular level, focusing on the pars intercerebralis (PI) neurons that photoperiodically change their neural activity and promote oviposition. Electrophysiological analyses showed that L-Glutamate acts as an inhibitory signal to PI neurons via glutamate-gated chloride channel (GluCl). Additionally, combination of electrophysiology and genetics revealed that knockdown of *got*, *gs*, and *glucl* disrupted cellular photoperiodic responses of the PI neurons, in addition to reproductive phenotypes. Our results reveal that the extracellular glutamate dynamics are photoperiodically regulated depending on the clock gene and play an essential role in the photoperiodic control of reproduction via inhibitory pathways.

## Introduction

To adapt to seasonal environmental changes, animals adequately modulate their physiological status and behavior according to each season. The photoperiod is an important environmental

**Data Availability Statement:** All relevant data are within the paper and its Supporting Information files.

**Funding:** This work was supported by the Ministry of Education, Culture, Sports, Science and

Technology-Japan Society for the Promotion of Science [Grants-in-Aid for Scientific Research JP20K15842 (to M.H.), https://kaken.nii.ac.jp/ja/grant/KAKENHI-PROJECT-20K15842/]. The funder had no role in study design, data collection and analysis, decision to publish, or preparation of the manuscript.

**Competing interests:** The authors have declared that no competing interests exist.

**Abbreviations:** CA, corpus allatum; CNS, central nervous system; ds, double-stranded; eaat2, *excitatory amino acid transporter 2*; GluCl, glutamate-gated chloride channel; gad, *glutamate decarboxylase*; got, *glutamate oxaloacetate transaminase*; gs, *glutamine synthetase*; MEM, minimum essential medium; NMDA, N-methyl-D-aspartate; PDF, pigment-dispersing factor; PI, pars intercerebralis; PL, pars lateralis; qPCR, quantitative PCR; RNAi, RNA interference; TTX, tetrodotoxin; vglut, *vesicular glutamate transporter*.

factor for sensing seasonal changes. In 1936, Bünning proposed a model in which endogenous circadian rhythms underlie the time measurement of photoperiod [1], and subsequent studies have examined the relationship between the circadian clock system and photoperiodic responses in both vertebrates and invertebrates [2–7]. Previous studies using insect models demonstrated that genetic suppression of expression of clock genes disrupted not only circadian rhythms but also photoperiodic responses of reproduction [8–14]. Thus, the molecular basis of the circadian clock is essential for photoperiodic responses. For seasonal adaptation, the photoperiodic information is processed based on the circadian clock system and transmitted to the central brain neurons that control photoperiodic traits in physiology and behavior. However, circadian clock-driven neural signals that convey photoperiodic information are poorly understood. Therefore, to understand regulatory mechanisms underlying the photoperiodic response, the manifestation of neural signals that respond to the photoperiod and convey photoperiodic information based on the circadian clock system is essential.

Here, we focused on glutamate, a classical neurotransmitter, as a neural signal for photoperiodic responses. Glutamate is a major neurotransmitter in the nervous system of animals, from the cephalochordate amphioxus (the invertebrate close to vertebrates) to mammals [15–17]. In mammals, glutamate primarily acts as a fast excitatory neurotransmitter and is used in approximately 50% of synaptic transmissions within the central nervous system (CNS) [17]. In insects, histological studies have demonstrated that glutamatergic neurons widely exist in the CNS [18–22]. Moreover, glutamatergic signals are involved in various physiological functions in insects, such as reproduction, circadian locomotor activity, sleep–wake balance, olfactory learning, long-term memory, and ON-OFF selectivity in visual systems [23–30]. However, despite the suggestion of widespread glutamate neurotransmission in the CNS, there have been no studies on the involvement of glutamate dynamics in photoperiodic responses based on the circadian clock.

In this study, we investigated the relationship between the brain extracellular glutamate dynamics and circadian clock-dependent photoperiodic responses in the bean bug *Riptortus pedestris*. This insect shows a clear photoperiodism in reproduction [31], and this photoperiodic response is disrupted by RNA interference (RNAi)-mediated suppression of different clock genes [8–10,32,33]. Furthermore, surgical experiments in *R. pedestris* have also demonstrated that the brain regions, pars intercerebralis (PI) and pars lateralis (PL), known as insect neuroendocrine centers, are involved in reproductive control [34,35]. Our recent study identified that large neurons in the PI photoperiodically change their neural activity based on the clock gene *period* (*per*) and contribute to promoting oviposition [32]. Taken together, it is hypothesized that photoperiodic information based on clock genes may be transmitted to reproductive control neurons, which contributes to photoperiodic modulations in reproduction in *R. pedestris*. Due to clarity of the photoperiodic responses and reproductive mechanism, we selected this insect as a good research model for analyzing the importance of glutamate dynamics in the photoperiodic regulation of reproduction.

First, we investigated extracellular glutamate levels in the whole brain and found that glutamate levels changed photoperiodically depending on the clock gene *per*. Second, we identified that the photoperiodic response in reproduction was attenuated by genetic modulation of the glutamate dynamics via RNAi-mediated knockdown of glutamate-metabolizing enzyme genes, *glutamate oxaloacetate transaminase* (*got*) and *glutamine synthetase* (*gs*). Additionally, electrophysiological and genetic analyses together revealed that L-Glutamate strongly inhibited the PI neuronal activity via a glutamate-gated chloride channel (GluCl) and that RNAi of *got*, *gs*, and *glucl* disrupted the photoperiodic changes in PI neuronal activity. These findings indicate the involvement of circadian clock-dependent glutamate dynamics in photoperiodic responses of both reproductive phenotypes and PI cellular activity.

## Results

### Extracellular glutamate levels in the whole brain photoperiodically change depending on the clock gene *period*

We compared extracellular glutamate levels in the brain between different day length conditions to examine the photoperiodic response of glutamatergic signals. First, we checked the photoperiodic response in the ovarian development of intact females. A majority of the intact females showed development of ovaries under long-day conditions (16 h light:8 h dark), whereas many females had immature ovaries under short-day conditions (12 h light:12 h dark) (S1 Fig). This indicated a clear photoperiodic response in reproduction. From these intact females, we measured the amount of brain extracellular glutamate by 24 h of brain culture (Fig 1A). Within both daytime (zeitgeber time, ZT3–5) and nighttime (ZT16–18) dissected groups, the extracellular glutamate concentration was significantly higher in females under short-day conditions than in those under long-day conditions (Fig 1B).

To examine whether the detected extracellular glutamate was physiologically released from brain cells, we additionally measured the extracellular glutamate concentration of the brain cultured in the medium supplemented with saline or tetrodotoxin (TTX), which is an inhibitor of $Na^+$ channel and subsequently inhibits the neurotransmitter release. Glutamate concentration in the TTX group was much lower than that in the saline group (Fig 1C). This result suggests that the measured extracellular glutamate is mainly due to neural activity-dependent release from brain cells rather than leaked from the brain due to injury to the brain.

A previous study in *R. pedestris* demonstrated that RNAi of the clock gene *per* abolished both circadian cuticle deposition rhythm and photoperiodic responses of reproduction [8]. This result suggests that gene knockdown of *per* attenuates circadian clock functions important for circadian rhythm and photoperiodism. Thus, we examined whether the clock gene *per* mediated photoperiodic changes in glutamate levels by double-stranded (ds) RNA-induced RNAi. The *per* expression levels in the brain were significantly lower in females injected with dsRNA for *per* (ds*per*) than those injected with control dsRNA for *β-lactamase* (ds*bla*) (S2 Fig). Similar to results reported previously [8,32,33], the photoperiodic response of ovarian development was disrupted in the ds*per*-injected females, whereas that in control ds*bla*-injected females was intact (S3 Fig).

We measured the extracellular glutamate levels in each dsRNA-injected female. Similar to intact females, the glutamate concentration was significantly higher under short-day conditions than that under long-day conditions in the control ds*bla*-injected group (Fig 1D). In contrast, in the ds*per*-injected group, glutamate concentration was low under both long-day and short-day conditions, and there was no significant difference in the glutamate concentration between long and short days (Fig 1D). These results indicate that photoperiodic change occurs in the extracellular glutamate levels within the brain and that it depends on expression of *per*.

### RNAi of glutamate-metabolizing enzyme genes disrupts photoperiodic responses of reproduction

Next, we examined whether extracellular glutamate dynamics were involved in the photoperiodic response of reproduction by RNAi of genes encoding glutamate-metabolizing enzymes, *got* and *gs* (S4 Fig). GOT catalyzes reversible transformation of aspartate and α-ketoglutarate to glutamate and oxaloacetate, and GS catalyzes the synthesis of glutamine from glutamate and ammonia. A previous study of the neuromuscular junctions in *Drosophila melanogaster* showed that loss-of-function mutations in *got* down-regulate synaptic glutamate levels and mutations in *gs* up-regulate glutamate levels [36]. Therefore, we first checked whether RNAi of

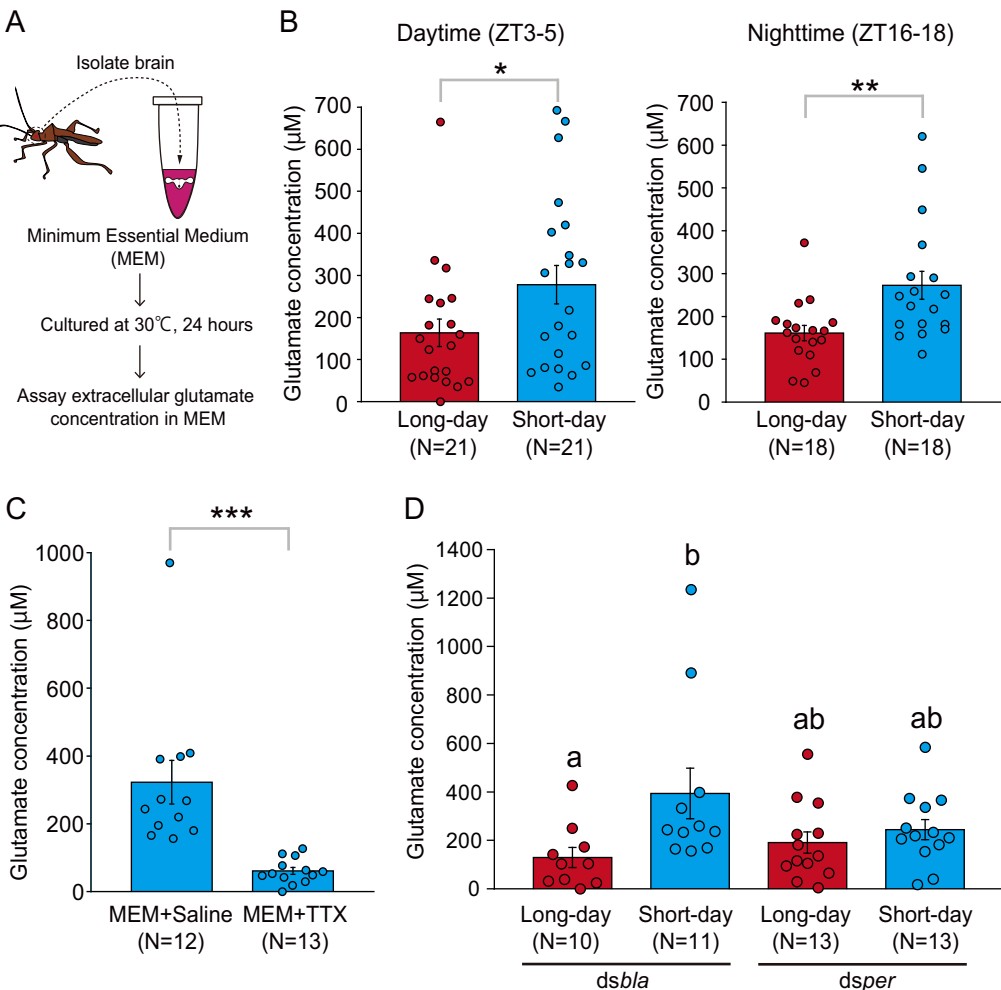

**Fig 1. Extracellular glutamate levels in the whole brain change photoperiodically depending on the clock gene**
***period.*** (A) Illustrations of measuring extracellular glutamate concentration by culturing the whole brain of *R. pedestris*.
(B) Glutamate concentration in intact females under long-day and short-day conditions. Females were dissected at
daytime (left, zeitgeber time, ZT3–5) and nighttime (right, ZT16–18). (C) Glutamate concentration in intact females
under short-day conditions cultured by the culture medium (MEM) with saline or MEM with 1 μM TTX (dissected at
ZT3–5). (D) Glutamate concentration in ds*bla*- and ds*per*-injected females under long-day and short-day conditions
(dissected at ZT3–5). B, C: Two-tailed Mann–Whitney *U* test, *: $P < 0.05$, **: $P < 0.01$, ***: $P < 0.001$, D: columns with
different letters indicate statistically significant differences (Steel–Dwass test, $P < 0.05$). Columns with error bars
indicate mean value ± SEM. The underlying data can be found in the S1 Data datasheet of numerical values for each fig.
xlsx. MEM, minimum essential medium; TTX, tetrodotoxin.

these enzyme-encoding genes can genetically manipulate extracellular glutamate levels under
long-day and short-day conditions in the group-bred females. In the control ds*bla* group,
there was a significant difference in glutamate concentration between long-day and short-day
conditions (Fig 2A), which was similar to intact females (Fig 1B). On the other hand, females
injected with dsRNA for *got* (ds*got*) showed low glutamate levels under both long days and
short days (Fig 2A). In contrast, glutamate concentration of dsRNA for *gs* (ds*gs*)-injected
females was high in both long-day and short-day conditions (Fig 2A). Within both ds*got* and
ds*gs* groups, there were no significant differences in glutamate concentration between long
days and short days. These results indicated that RNAi of *got* induced a decrease in the brain

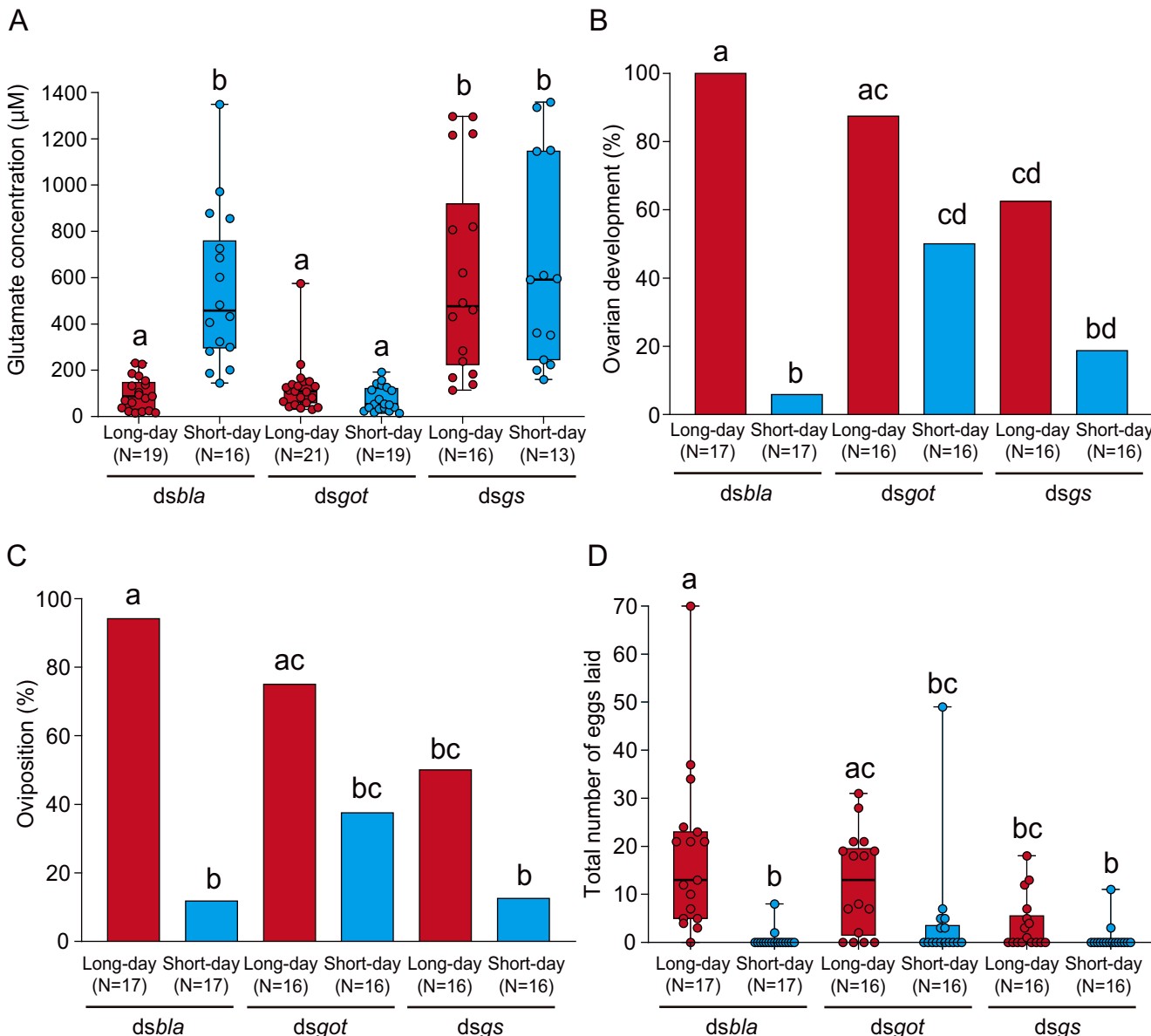

**Fig 2. RNAi of *got* and *gs* affects extracellular glutamate levels and attenuates photoperiodic responses in reproduction.** (A) Box and scatter plots showing the glutamate concentration in ds*bla*-, ds*got*-, and ds*gs*-injected females under long-day and short-day conditions. (B, C) Columns showing proportions of (B) ovarian development and (C) oviposition in ds*bla*-, ds*got*-, and ds*gs*-injected females under long-day and short-day conditions. (D) Box and scatter plots showing the total number of eggs laid in each group. (A–D) Columns and box plots with different letters indicate statistically significant differences (A, D: Steel–Dwass test; B, C: Tukey-type multiple comparisons for proportions, $P < 0.05$). (A, D) Lines at the top, middle, and bottom of the box plots indicate the upper quartile, median, and lower quartile, respectively. Upper and lower whiskers of the box plots indicate the maximum and minimum values, respectively. The underlying data can be found in the S1 Data datasheet of numerical values for each fig.xlsx. *got*, *glutamate oxaloacetate transaminase*; *gs*, *glutamine synthetase*; RNAi, RNA interference.

glutamate levels under short-day conditions, and RNAi of *gs* increased glutamate levels under long-day conditions.

Further, we examined the effects of RNAi of *got* and *gs* on the photoperiodic responses of reproduction in individually bred females. In the control ds*bla* group, a majority of the females showed development of ovaries and oviposited under long-day conditions, but not under short-day conditions, suggesting clear photoperiodic responses (Fig 2B–2D). However, in the

ds*got*-injected group, in which glutamate levels were down-regulated, approximately 40% to 50% of the females showed development of ovaries and oviposited even under diapause-inducing short-day conditions (Fig 2B–2D). In contrast to the ds*got* group, in ds*gs*-injected females, in which glutamate levels were up-regulated, only approximately 50% to 60% of the females showed development of ovaries and oviposited even under non-diapause–inducing long-day conditions (Fig 2B–2D). In both ds*got*- and ds*gs*-injected females, there were no significant differences in reproductive phenotypes between long-day and short-day conditions (Fig 2B–2D). Finally, we examined the knockdown specificity of dsRNA-induced RNAi. ds*got* and ds*gs* specifically reduced the mRNA expression of target genes (*got* and *gs*, respectively) under both long-day and short-day conditions (S5 Fig). On the other hand, there were no significant differences in expression of *got* and *gs* between long-day and short-day conditions within each dsRNA-injected group (S5 Fig). These findings clarify that RNAi-mediated knockdown of *got* decreases extracellular glutamate concentration that may avert reproductive diapause under short-day conditions, and RNAi of *gs* increases glutamate concentration that may induce diapause under long-day conditions.

## RNAi of *got* and *gs* disrupts the photoperiodic neural response in the PI neurons

In *R. pedestris*, large PI neurons show *per*-dependent photoperiodic changes in their firing activity and play an important role in promoting oviposition [32]. Thus, we examined whether the glutamate dynamics are involved in the photoperiodic control of PI neuronal activity by RNAi and electrophysiology analyses. Similar to a previous study [32], large PI neurons showed various firing activities, and we classified them into 3 types: high-frequency burst, non-burst, and silent (Fig 3A). Similar to results of the previous study [32], PI neurons in the control ds*bla* group showed a clear photoperiodic response in neuronal activity; many PI neurons showed high-frequency firing under long-day conditions, whereas most PI neurons were silent under short-day conditions (Fig 3B–3D). Within the ds*bla* group, there were significant differences in firing proportions, instantaneous frequency, and firing number between long- and short-day conditions (Fig 3B–3D). In contrast, in ds*got*- and ds*gs*-injected females, approximately 40% to 60% of the PI neurons showed spontaneous firing under both long-day and short-day conditions, and there was no significant difference in firing proportions between long- and short-day conditions (Fig 3B). Additionally, within the ds*got* and ds*gs* groups, the instantaneous frequency and firing number were also not significantly different between long- and short-day conditions (Fig 3C and 3D). These results indicate that RNAi of glutamate-metabolizing enzyme genes, *got* and *gs*, attenuates the photoperiodic responses not only in the reproductive phenotypes but also in neural activity of PI neurons.

## L-Glutamate strongly and acutely inhibits firing activity of PI neurons via GluCl

Next, we investigated the direct effects of glutamate on PI neuronal activity by perfusion experiments. Perfusion of L-Glutamate strongly and acutely suppressed spontaneous firing of PI neurons (Fig 4A). The firing activity completely disappeared during the perfusion of L-Glutamate, and the activity was restored after wash-out (Fig 4B–4D). On the other hand, perfusion of D-Glutamate, an enantiomer of L-Glutamate, did not have significant effects on the neural activity of PI neurons, whereas slightly attenuated the neural activity in some cells (S6 Fig). These results clarify that L-Glutamate, but not D-Glutamate, has a strong inhibitory effect on the PI neurons.

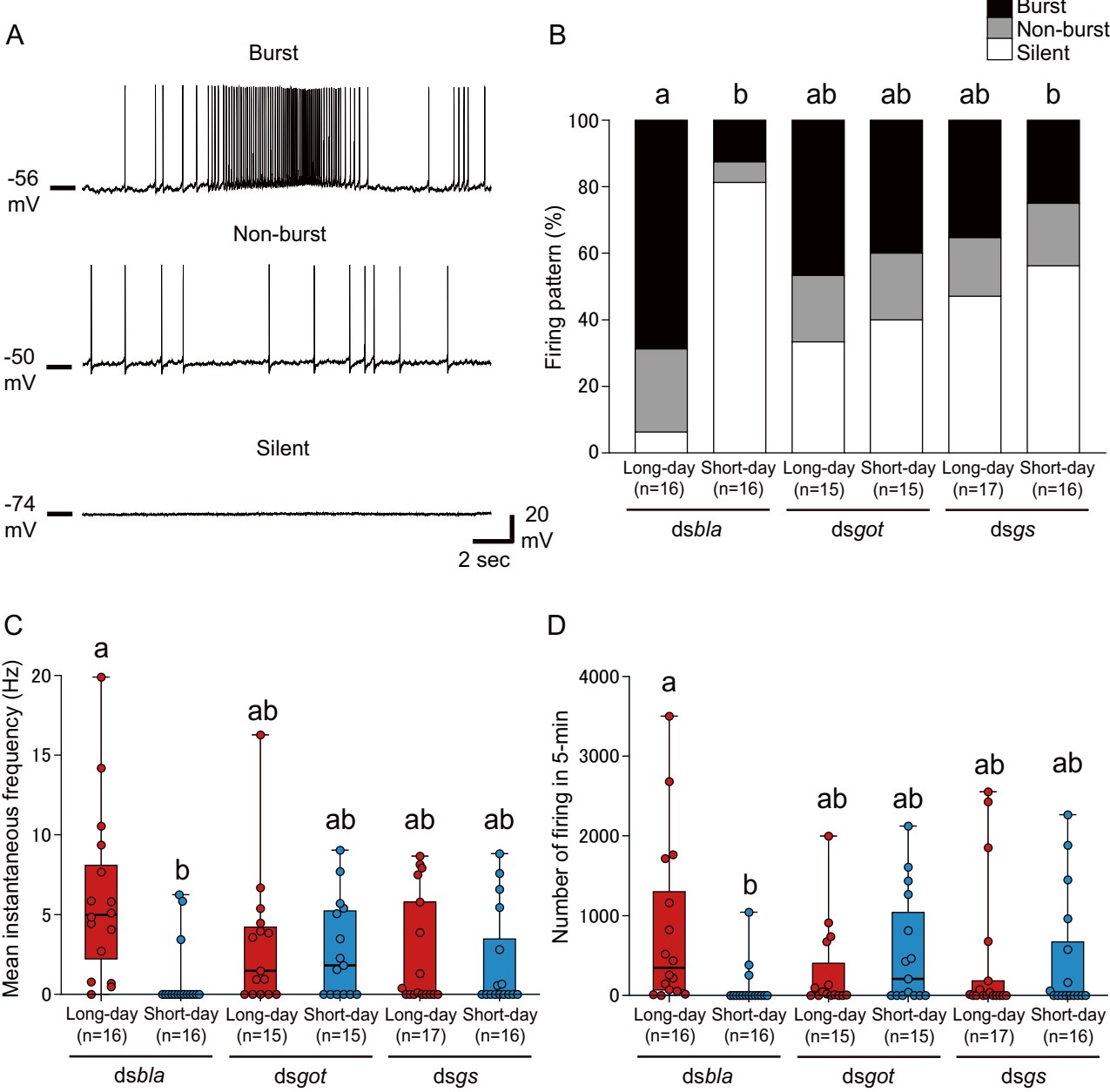

**Fig 3. RNAi of *got* and *gs* disrupts photoperiodic responses in large PI neurons.** (A) Representative traces showing burst (upper), non-burst (middle), and silent (lower) patterns in PI neurons. (B) Columns showing proportions of the 3 firing patterns in ds*bla*-, ds*got*-, and ds*gs*-injected females under long-day and short-day conditions. (C, D) Box and scatter plots showing the (C) mean instantaneous frequency and (D) number of firing events in 5 min in each group. (B–D) Columns and box plots with different letters indicate statistically significant differences (B: Tukey-type multiple comparisons for proportions ["silent" rate comparison]; C, D: Steel–Dwass test, $P < 0.05$). (C, D) Lines at the top, middle, and bottom of the box plots indicate the upper quartile, median, and lower quartile, respectively. Upper and lower whiskers of the box plots indicate the maximum and minimum values, respectively. The underlying data can be found in the S1 Data datasheet of numerical values for each fig.xlsx. PI, pars intercerebralis; RNAi, RNA interference.

Further, we examined the receptors mediating the inhibitory effects of L-Glutamate. In invertebrates, GluCl functions as an inhibitory ionotropic glutamate receptor [37]. When glutamate binds to GluCl, GluCl immediately suppresses neural activity by opening a channel that passes chloride ions ($Cl^-$). Thus, GluCl is a prime candidate for acute glutamate-induced

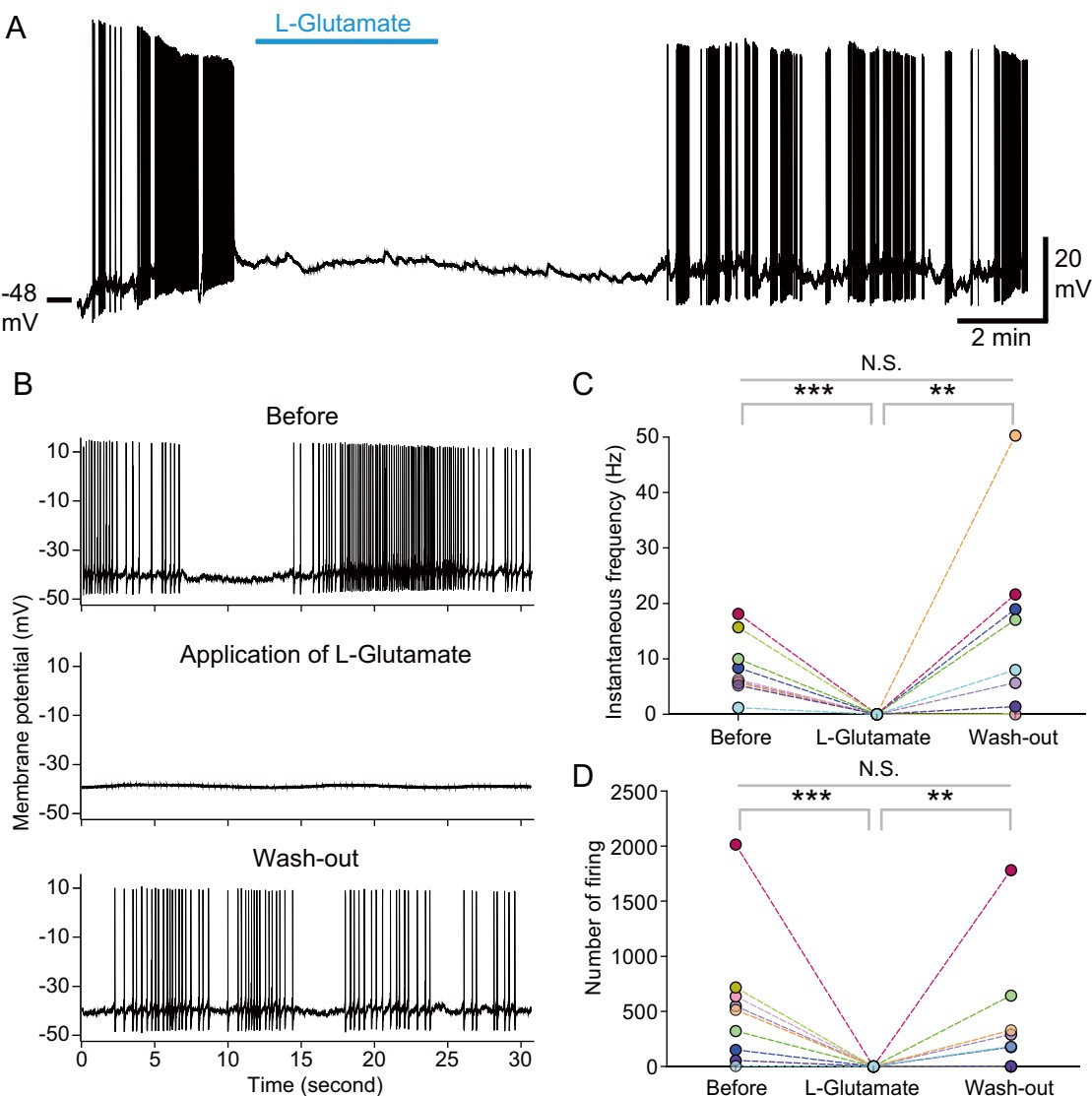

**Fig 4. Perfusion of L-Glutamate acutely and strongly inhibits the spontaneous firing activity of PI neurons.** (A) A representative trace showing effects of 1 mM of L-Glutamate perfusion on the spontaneous firing activity of PI neurons. (B) Traces showing the spontaneous activity of a PI neuron before the perfusion of glutamate (upper), during application of glutamate (middle), and after wash-out (lower). (C, D) Line graphs showing the (C) instantaneous frequency and (D) number of firing events in 3 min of "Before," "L-Glutamate," and "Wash-out" within each PI cell ($n = 9$). We performed repeated Friedman test with post hoc Steel–Dwass test. Repeated Friedman test shows statistically significant differences in the instantaneous frequency and number of firing events ($P < 0.01$). Symbols show statistical $P$ values by Steel–Dwass test (**: $P < 0.01$, ***: $P < 0.001$, N.S.: not significant). The underlying data can be found in the S1 Data datasheet of numerical values for each fig.xlsx. PI, pars intercerebralis.

neural inhibition. To examine the involvement of GluCl in glutamate-induced suppression of PI neurons, we analyzed glutamate-induced current properties using voltage clamp method. Under normal recording conditions, the intracellular pipette solution contained low chloride ions (7 mM) and extracellular solution contained high chloride ions (147 mM) (Fig 5A). Under these conditions, L-Glutamate perfusion induced small inward currents at −80 mV holding, small outward currents at −60 mV, and large outward currents at more depolarized states (−40 mV and −20 mV) (Fig 5A). The reversal potential of L-Glutamate-induced currents was approximately −66.3 mV as per the current–voltage curve (Fig 5C), which was close to the

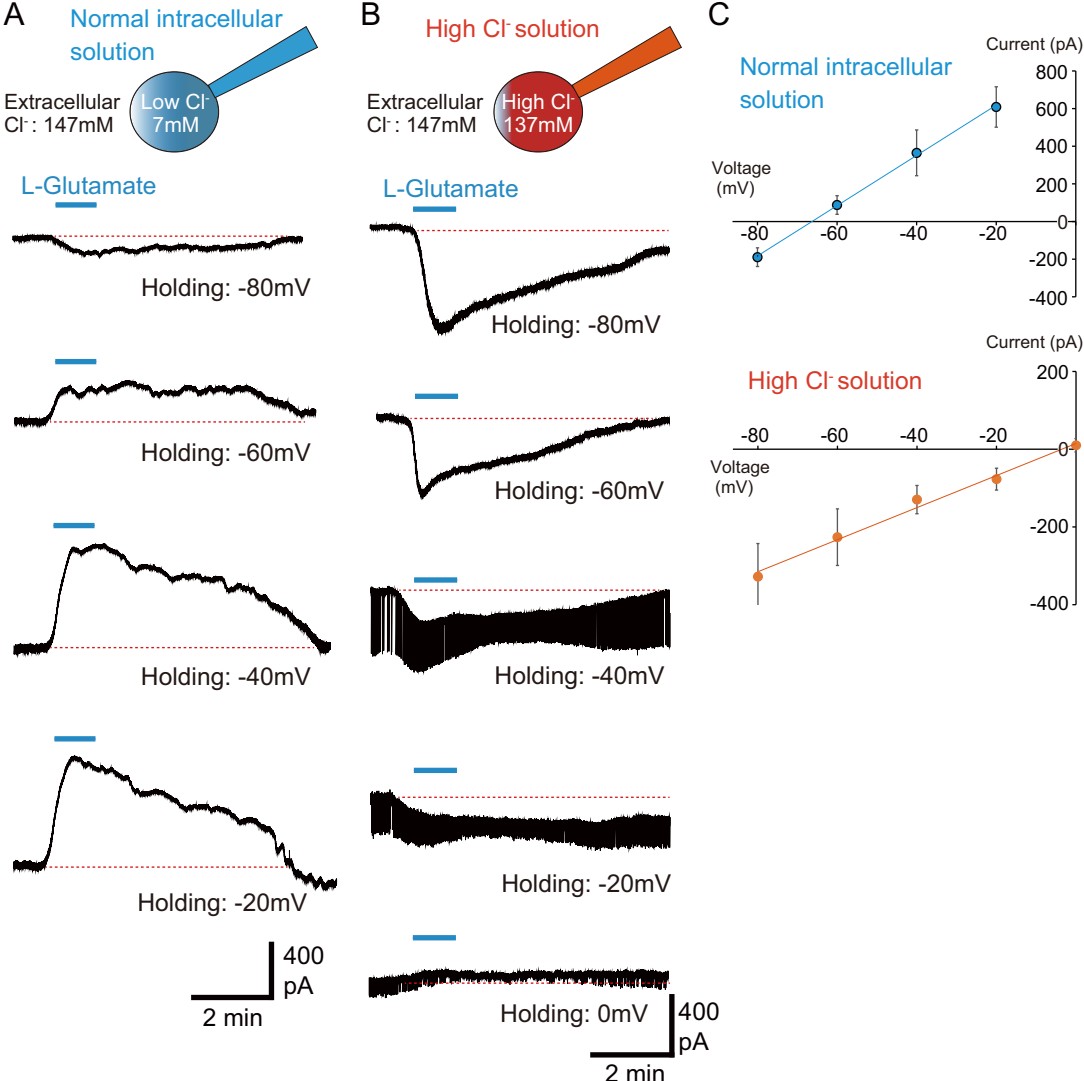

**Fig 5. L-Glutamate induced currents in PI neurons are mediated by chloride ion.** (A, B) Representative traces showing 1 mM of L-Glutamate-induced currents at each holding potential using (A) normal intracellular solution and (B) high intracellular Cl⁻ solution. (C) Graphs showing the current–voltage curves of glutamate-induced currents in the normal intracellular solution (upper, $n = 5$) and high Cl⁻ solution (lower, $n = 5$). Plots with error bars indicate mean value ± SEM. The underlying data can be found in the S1 Data datasheet of numerical values for each fig.xlsx. PI, pars intercerebralis.

theoretical equilibrium potential of chloride ions (−72.7 mV). Next, we analyzed the L-Glutamate-induced currents by changing the intracellular conditions to a high intercellular Cl⁻ concentration (137 mM) (Fig 5B). Due to the high Cl⁻ internal concentration, L-Glutamate perfusion induced large inward currents at −80 mV (Fig 5B). L-Glutamate-induced inward currents also occurred at −60 mV to −20 mV holdings (Fig 5B), whereas outward currents were evoked under the normal intracellular solution (Fig 5A). At the high Cl⁻ internal concentration, the reversal potential was approximately −3.6 mV (Fig 5C), which was close to the equilibrium potential of chloride ions (−1.7 mV). These results strongly indicate that the L-Glutamate-induced effect in PI neurons is mediated by chloride ion current.

To examine whether GluCl mediates the L-Glutamate-induced currents in the PI, we also performed the co-perfusion experiment of L-Glutamate and the antagonist for GluCl,

picrotoxin [38,39], under the normal recording conditions. We first perfused only L-Glutamate to the PI cell and found that single perfusion of L-Glutamate evoked large outward currents at −20 mV (Fig 6A and 6B). Then, we co-perfused L-Glutamate and picrotoxin and found that L-Glutamate-induced currents were dramatically attenuated under picrotoxin (Fig 6A and 6B). After wash-out, outward currents by the single perfusion of L-Glutamate were restored to some extent (Fig 6A and 6B). Additionally, we confirmed the expression of *glucl* in PI neurons using single-cell PCR method [32]. We detected predicted *glucl* gene sequences (isoform1, 2) in the RNA sequence data (S7 Fig). Eight PI cells from 5 females (*n* = 40 cells) were collected, and expression of *glucl* was examined by nested PCR (primers of *glucl* targeted the common sequence of 2 isoforms). The housekeeping gene *beta-tubulin* (*tubulin*) was expressed in all PI cells (positive control; Fig 6C and S1 Table). We also found that a majority of the PI cells (*n* = 36 cells within 40 cells) expressed *glucl* (Fig 6C and S1 Table). Thus, results of electrophysiology and PCR clearly suggest that glutamate directly suppresses PI neural activity via GluCl.

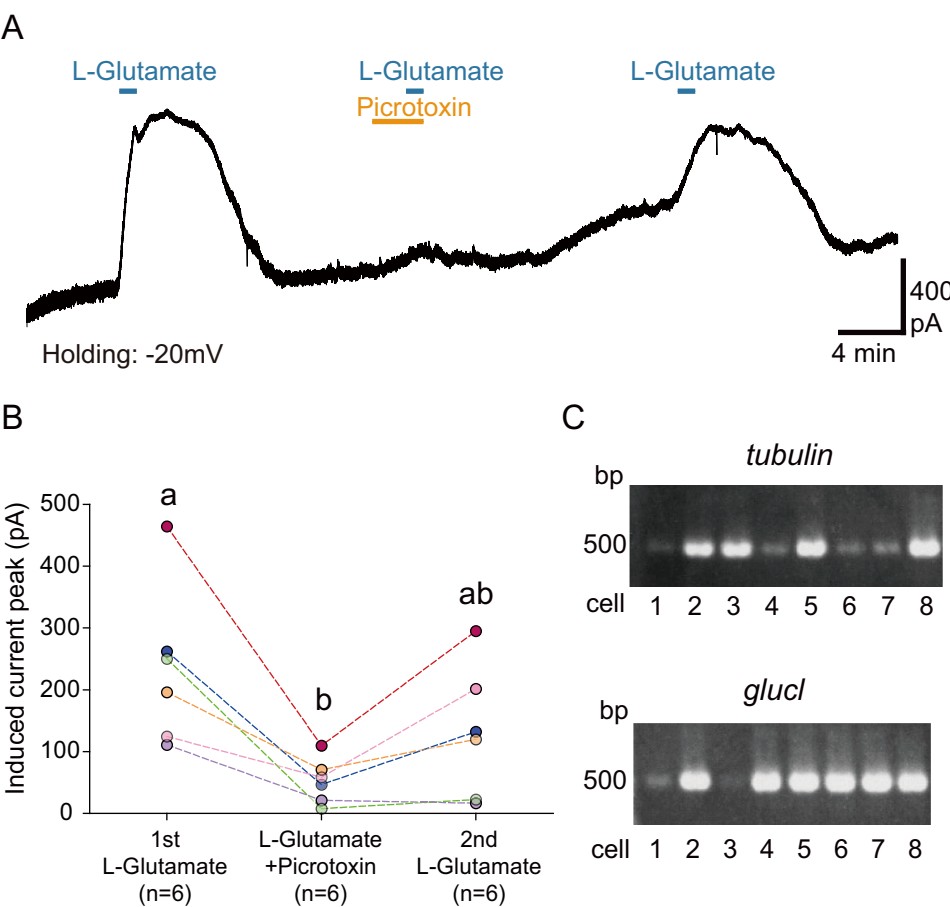

**Fig 6. GluCl mediates glutamate-induced currents in PI neurons.** (A) A representative trace showing induced currents by single perfusion of 300 μM L-Glutamate and co-perfusion of 300 μM L-Glutamate with 100 μM picrotoxin at −20 mV holding under the normal intracellular condition. (B) Line graphs showing the induced current peak by first single perfusion of L-Glutamate (1st L-Glutamate), co-perfusion of L-Glutamate and picrotoxin (L-Glutamate + Picrotoxin), second single perfusion of L-Glutamate (2nd L-Glutamate). Plots with different letters indicate statistically significant differences (one-way ANOVA with post hoc Tukey–Kramer test, *P* < 0.05). (C) Representative images showing expression of *tubulin* and *glucl* in 8 PI cells from a single female ("Female d" in S1 Table). The underlying data can be found in the S1 Data datasheet of numerical values for each fig.xlsx. GluCl, glutamate-gated chloride channel; PI, pars intercerebralis.

## RNAi of *glucl* diminishes photoperiodic responses of reproductive phenotypes and PI neuronal activity

We also examined whether GluCl mediates photoperiodic responses of reproductive functions and PI neuronal activity using RNAi. dsRNA for *glucl* (ds*glucl*) significantly reduced the expression of *glucl* in both photoperiods than control ds*bla* (S8 Fig). Similar to results shown in Fig 2, ds*bla* females showed clear photoperiodic changes in reproduction (Fig 7A–7C). However, a majority of the ds*glucl*-injected females showed development of ovaries and oviposited under both long-day and short-day conditions, and there were no significant differences (Fig 7A–7C). The analyses highlighted importance of GluCl in the photoperiodic control of reproduction.

We also recorded the neuronal activity of the PI neurons in these females. In the ds*bla* females, many PI neurons showed high-frequency burst firing under long-day conditions, whereas most PI neurons were silent under short-day conditions (Fig 7D). There were significant differences in the instantaneous frequency and number of firing between long- and short-day conditions in control ds*bla* females (Fig 7E and 7F). In contrast, in the ds*glucl* group, approximately 60% of the PI neurons showed spontaneous firing both under long- and short-day conditions, and there were no significant differences in firing proportions, instantaneous frequency, and number of firing events (Fig 7D–7F). Taken together, the electrophysiological results suggested that GluCl is required for the photoperiodic response in PI neurons.

## Discussion

Advanced genetic analyses using insect models have suggested that circadian clock genes are involved in photoperiodism [8–13,33,40–42]. However, neural signaling pathways that convey the photoperiodic information based on clock genes remain largely unknown. In the present study, we focused on glutamatergic signaling and identified that extracellular glutamate levels photoperiodically change in the brain of *R. pedestris*, depending on the expression of clock gene *period*. Next, to investigate the involvement of glutamatergic signals in photoperiodism, we performed RNAi-mediated knockdown of glutamate-metabolizing enzyme genes. Because complete loss-of-function mutations of glutamate signals may have critical effects on insects, partial gene knockdown by RNAi was suitable for the present study. Using a combination of electrophysiology and genetic analyses, we demonstrated that RNAi-mediated suppression of gene expression of glutamate-metabolizing enzymes and an inhibitory glutamate receptor disrupted the photoperiodic responses of reproductive phenotypes and PI neuronal activity. These findings suggest the significance of extracellular glutamate dynamics in the clock gene-dependent photoperiodic control of reproduction.

Because RNAi of the clock gene *period* abolished changes in glutamate dynamics important for photoperiodic reproductive regulation, the glutamatergic signal may be involved in photoperiodic signal transduction from clock cells to reproductive control cells. The present photoperiodic change in whole brain-glutamate levels may be correlated with changes in glutamatergic transmission of this pathway. In this study, we used ds*got* and ds*gs* as genetic tools to manipulate the glutamate dynamics. On the other hand, the mRNA levels of *got* and *gs* in the whole head did not significantly change according to photoperiod (S5 Fig), suggesting that other glutamate-related factors are involved in the photoperiodic change in glutamate dynamics. We additionally checked brain mRNA expression of glutamate-converting enzyme, *glutamate decarboxylase* (*gad*), and glutamate transporters, *excitatory amino acid transporter 2* (*eaat2*) and *vesicular glutamate transporter* (*vglut*) (S9 Fig), which were reported to modulate the glutamate levels [36,43,44]. mRNA expression of *vglut*, but not other glutamate-related genes, was statistically higher under short days than long days, but difference was not drastic

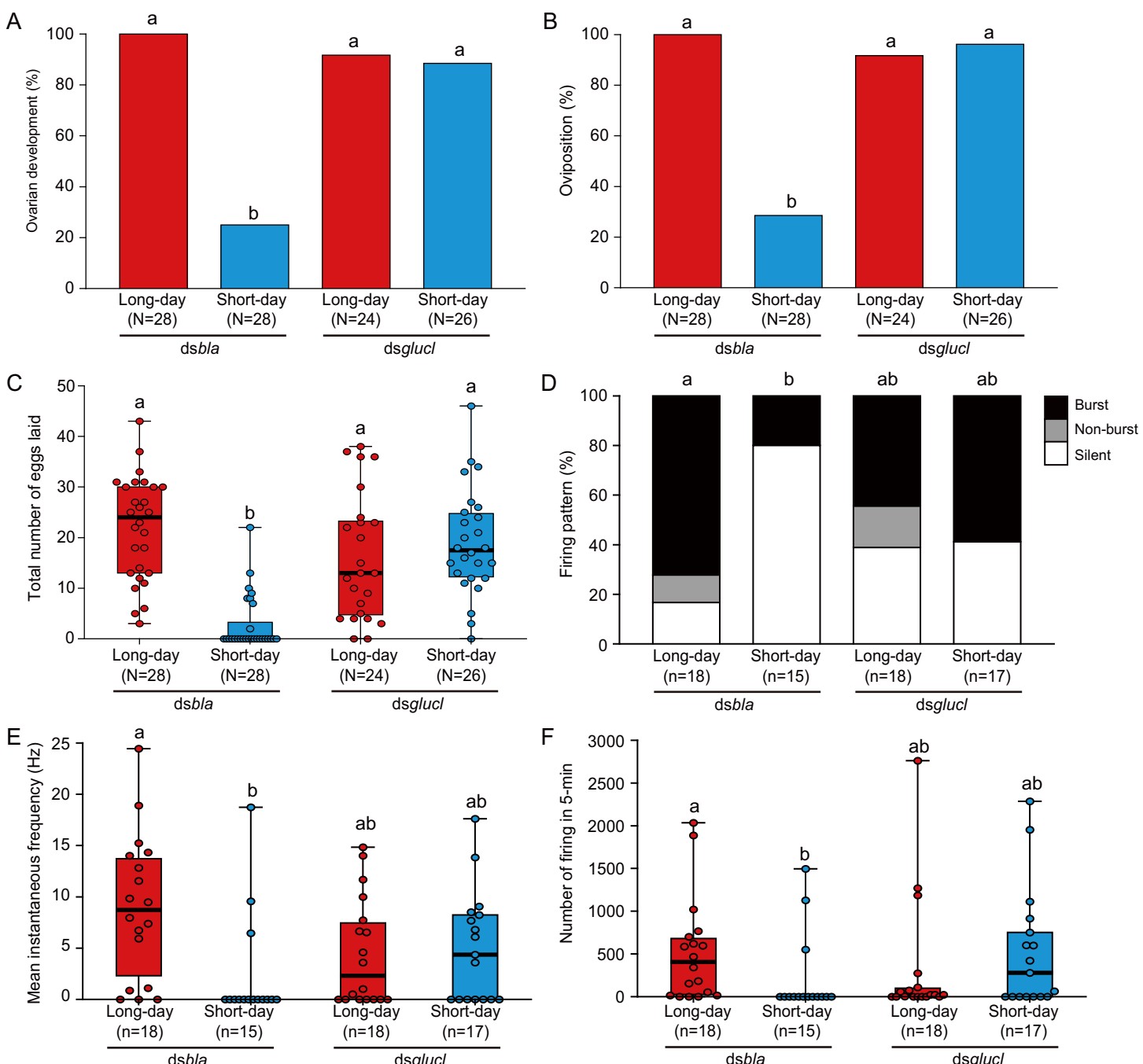

**Fig 7. RNAi of *glucl* disrupts photoperiodic responses in the reproductive phenotypes and PI neuronal activity.** (A, B) Columns showing proportions of (A) ovarian development and (B) oviposition of ds*bla*- and ds*glucl*-injected females under long-day and short-day conditions. (C) Box and scatter plots showing total number of eggs laid in each group. (D) Columns showing proportions of the 3 firing patterns in ds*bla*- and ds*glucl*-injected females under long-day and short-day conditions. (E, F) Box and scatter plots showing the (E) mean instantaneous frequency and (F) number of firing events in 5 min in each group. Columns and box plots with different letters indicate statistically significant differences (A, B, D: Tukey-type multiple comparisons for proportions [D: "silent" rate comparison]; C, E, F: Steel–Dwass test, $P < 0.05$). (C, E, F) Lines at the top, middle, and bottom of the box plots indicate the upper quartile, median, and lower quartile, respectively. Upper and lower whiskers of the box plots indicate the maximum and minimum values, respectively. The underlying data can be found in the S1 Data datasheet of numerical values for each fig.xlsx. PI, pars intercerebralis; RNAi, RNA interference.

(S10 Fig). In *R. pedestris*, another group recently reported that gene knockdown of *vglut* tended to abort the reproductive diapause [45]. Thus, there is a possibility that VGLUT contributes to the photoperiodic regulation of reproduction thorough control of extracellular glutamate levels. In future work, it is necessary to analyze carefully the mechanism that photoperiodically controls extracellular glutamate dynamics.

Although we have not yet identified clock cells located upstream of the glutamatergic signal, some dorsal clock gene-expressing cells in the brain themselves are glutamatergic in *D. melanogaster* [23]. In *R. pedestris*, 4–16 PERIOD-immunoreactive cells are found in the dorsal regions of the protocerebrum [46]. Thus, these dorsal PERIOD cells in *R. pedestris* might be glutamatergic and one of candidate clock cells involved in the glutamatergic photoperiodic transmission. Additionally, in *R. pedestris*, 2 PERIOD-immunoreactive cells are also found close to the pigment-dispersing factor (PDF) immunoreactive cells at the anterior base of the medulla [46]. Ablation of the anterior base of the medulla containing PDF immunoreactive cells disrupts the photoperiodic response of reproduction in *R. pedestris* [47]. Therefore, PERIOD cells at the anterior base of the medulla are also candidates of clock cells for assessing photoperiodic responses. Specific ablations targeting clock cells will help identify cells that play a central role in the glutamatergic photoperiodic transmission.

To the best of our knowledge, there has been no study that correlated extracellular glutamate levels and photoperiodic responses, although some studies have reported the involvement of glutamate in insect reproductive control. Injection of an antagonist for N-methyl-D-aspartate (NMDA) glutamate receptor significantly decreased the number of laid eggs in *Bicyclus anynana* (a butterfly) and *Gryllus bimaculatus* (a cricket) [30]. Some ionotropic glutamate receptors, such as GluCl and NMDA, have been reported to mediate the biosynthesis of juvenile hormones that play an important role in insect reproduction in the corpus allatum (CA) of *Diploptera punctata* (a cockroach) [48–50]. Therefore, modulation of glutamatergic signals may contribute to the photoperiodic control of reproduction in various insect species in addition to *R. pedestris*.

Since the photoperiodic change of glutamate levels can be detected at whole-brain levels, it is considered that the glutamatergic signal conveys photoperiodic information to multiple reproductive control cells. In *R. pedestris*, the oviposition-promoting PI neurons photoperiodically change their spontaneous neural activity depending on *per* expression, which suggests that the PI neurons are one of strong candidates receiving photoperiodic information from the circadian clock [32]. Thus, we focused on PI neurons as one of the reproductive control cells that receive glutamatergic signals from clock cells. We found that glutamate strongly inhibits neuronal activity of PI neurons via an inhibitory ionotropic receptor GluCl and that RNAi of *got*, *gs*, and *glucl* disrupted the photoperiodic responses of PI neuronal activity. Therefore, PI neurons may mediate glutamatergic modulation of oviposition in *R. pedestris* according to the photoperiod. PI, an insect endocrine center, is developmentally homologous to the hypothalamus in vertebrates [51]. Surgical ablation and genetic silencing of PI neurons disrupts the photoperiodic control of reproduction in the flies *Protophormia terraenovae* and *D. melanogaster* [52,53]. Also in *Plautia stali* (a bug), the PI neurons photoperiodically change their neuronal activity [54]. Based on these studies and results of the present study, we assume that PI may receive photoperiodic information and serve as an output center for insect photoperiodic control of reproduction.

In contrast, although RNAi of *got*, *gs*, and *glucl* attenuated photoperiodic response in ovarian development, ablation of PI did not disrupt photoperiodic response in *R. pedestris* [35]. Thus, the glutamatergic signal may also convey photoperiodic information to other regulatory pathways for the photoperiodic control of ovarian development. The brain region PL may be a candidate for glutamate-mediated control of ovarian development. The PL is the brain region

wherein various neuroendocrine regulatory cells exist similar to the PI [55] and plays an essential role in the induction of diapause under short-day conditions in some insects, including *R. pedestris* [35,52,56]. Additionally, there is a possibility that CA, an essential endocrine organ for insect reproductive control, directly receive glutamatergic inputs. Glutamatergic neurons directly project into the CA in *D. punctata* [49], and applications of glutamate and glutamate receptor agonists evoke electrophysiological and $Ca^{2+}$ responses of the CA cells in *D. punctata* and *G. bimaculatus* [48–50,57]. Based on these reports, we hypothesize that the glutamatergic signal parallelly regulates the brain neuroendocrine centers, PI and PL, and the endocrine organ CA that comprehensively contributes to the photoperiodic regulation of reproduction (Fig 8).

ds*got* and ds*gs* adversely affected extracellular glutamate levels (ds*got* induced low glutamate, ds*gs* induced high glutamate levels). On the other hand, reproductive phenotypes and PI cell activities in ds*got* and ds*gs* groups seemed to be intermediate between those in the control ds*bla* under long-day and short-day conditions, whereas ds*got* and ds*gs* tended to induce the

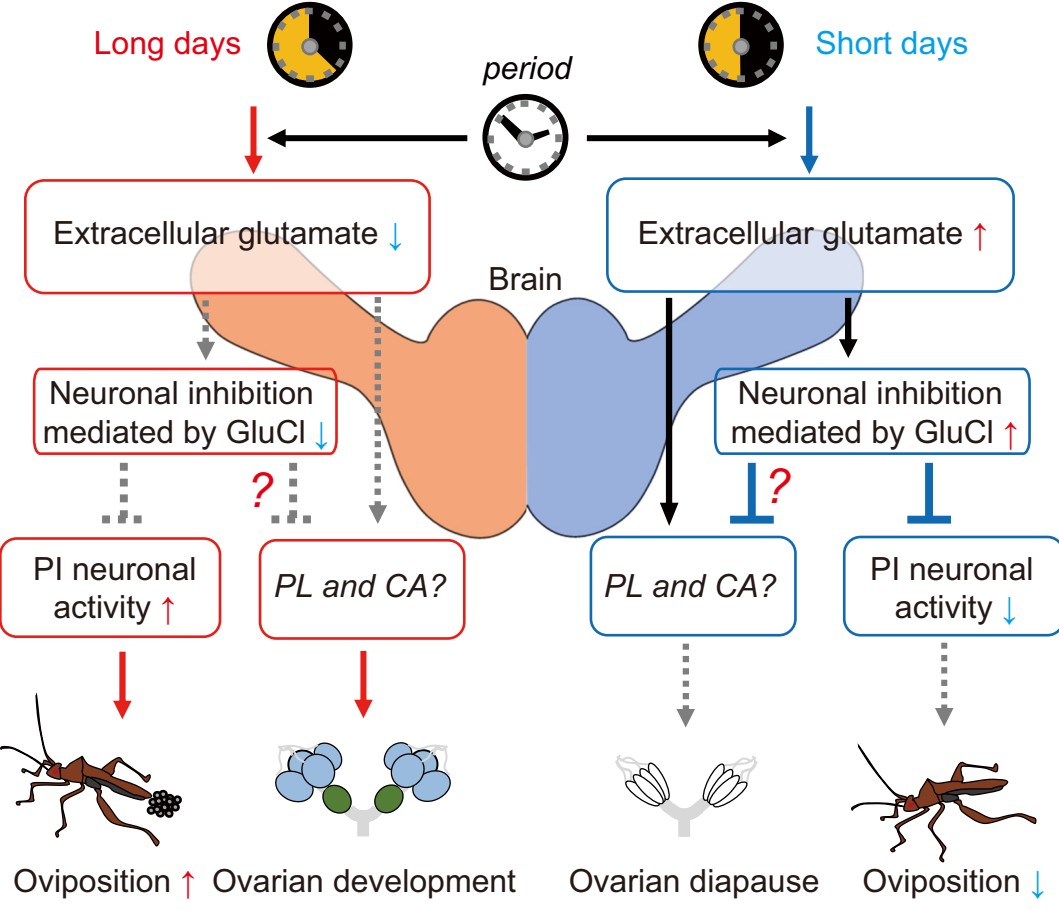

**Fig 8. Schematic illustration shows predicted hierarchical glutamatergic pathway for reproductive control.** Brain glutamate levels photoperiodically change depending on the clock gene *period*. A glutamatergic signal photoperiodically inhibits the PI neuronal activity via GluCl, which contributes to control of oviposition. The glutamatergic signal may parallelly regulate neuroendocrine centers (such as PL and CA) for photoperiodic regulations of ovarian development. CA, corpus allatum; GluCl, glutamate-gated chloride channel; PI, pars intercerebralis; PL, pars lateralis.

long-day and short-day phenotypes, respectively. The previous study in *P. terraenovae* demonstrated that ablation of PDF-immunoreactive clock cells induced the rate of ovarian diapause to about 50% under both long-day and short-day conditions, which may result from the inability of photoperiodic time measurement [58]. Therefore, we currently assume that the glutamatergic signal is also involved in the photoperiodic time measurement in clock cells, and ds*got* and ds*gs* may have disrupt the photoperiodic time measurement, which induced the intermediate reproductive phenotypes and PI activities.

In summary, the present analyses revealed *period*-dependent photoperiodic changes in extracellular glutamate levels and the significance of inhibitory glutamatergic signals in photoperiodic responses. Glutamate, which is up-regulated under short-day conditions depending on expression of *period*, may act as an essential neural molecule that induces diapause phenotypes via different neurosecretory pathways, including PI neurons. Glutamate is a major neurotransmitter in both vertebrates and invertebrates, but little attention has been paid to its involvement in the photoperiodic regulation. Future research focusing on the involvement of glutamatergic dynamics in photoperiodic responses may contribute to improved understanding of the brain neural pathways that convey photoperiodic information based on the circadian clock.

## Materials and methods

### Insects

*R. pedestris* was collected from Machikaneyama (36˚48′N, 135˚27′E, Toyonaka, Osaka, Japan) and maintained for 14 to 18 generations. The bugs were bred on soybeans, red clover seeds, and water supplemented with L-ascorbic acid sodium salt (0.05%) and L-cysteine (0.025%). Before eclosion, insects were kept under short-day conditions (12 h light/12 h dark) at 25 ± 1.5˚C. After eclosion, *R. pedestris* were bred under short-day or long-day conditions (16 h light/8 h dark) at 25 ± 1.5˚C. For glutamate assay and glutamate perfusion experiments, multiple females (approximately 8 to 16 females per cup) were maintained in a large plastic cup (diameter: 15.5 cm, depth: 9 cm). For the other experiments, after eclosion, females were individually maintained in a small plastic cup (diameter: 10 cm, depth: 4 cm). We classified the ovarian stage of each female from 0 to V; females with stages 0, I, and II ovaries were considered to have reproductive diapause and those with stages III, IV, and V ovaries to have non-diapause [31]. In individual rearing, we counted the total number of eggs laid 20 to 22 days after eclosion.

### dsRNA-induced RNA interference

dsRNA-induced RNAi experiments were performed as described in a previous study by our group [32]. We extracted and purified template RNA from whole heads of female *R. pedestris* 20 days after eclosion by using the FastGene RNA Basic Kit with DNase-I (Nippon Genetics, Tokyo, Japan). DNA fragments containing the T7 promoter were amplified using PrimeScript RT-PCR Kit (Takara Bio, Shiga, Japan), and dsRNAs were synthesized based on the DNA fragments using T7 RiboMAX Express RNAi System (Promega, Madison, Wisconsin, United States of America). For a working dsRNA solution, the concentration of dsRNA was adjusted to 1 μg/μL with pure water. Within 24 h of eclosion, 1 μL of dsRNA working solution (1 μg/μL) was injected into the head of insect. Primer sets for making dsRNAs are listed in S2 Table.

### Quantitative PCR

To examine the effects of RNAi on expression of *per*, total RNA from the whole brain was extracted 3 days after eclosion under the short-day conditions at zeitgeber time (ZT, the start

time of the light period corresponds to ZT0) 1–2 and purified using FastGene RNA Basic Kit with DNase-I (Nippon Genetics, Tokyo, Japan). To analyze the effects of RNAi on expression of *got*, *gs*, *glucl*, total RNA from the whole head was extracted 20 to 22 days after eclosion at ZT3–5 by the same kit. To examine photoperiodic expression changes of glutamate-related genes, total RNA from the whole brain of intact females under long or short days was extracted 20 to 21 days after eclosion at ZT3–5 by the same kit. Based on the total RNA, we synthesized cDNA using a reverse transcription kit (FastGene cDNA Synthesis 5× ReadyMix, Nippon Genetics, Tokyo, Japan). We performed quantitative PCR (qPCR) using an Applied Biosystems 7500 Real-Time PCR System (Thermo Fisher Scientific, Waltham, Massachusetts, USA) with FastStart Universal SYBR Green Master Rox (F. Hoffmann-La Roche, Basel, Switzerland). mRNA expression of each gene was quantified by the standard curve method. The expression level was normalized by a reference gene *tubulin* [59]. Primer sets for qPCR are listed in S2 Table.

## Glutamate assay in cultured brains

We measured glutamate levels in the brains of intact and dsRNA-injected females. Twenty to 22 days after eclosion, the bugs were anesthetized on ice and mounted in clay. The whole brains were carefully unsheathed and removed at ZT3–5 except the nighttime (ZT16–18) measurement of intact for investigating a time-of-day dependent change (Fig 1B). A single whole brain was cultured in 30 μL of minimum essential medium with Hank's salts with L-glutamine without sodium bicarbonate (MEM, Thermo Fisher Scientific-Gibco, Massachusetts, USA) and containing 20 mM HEPES and 5 ppm Tween 80 (pH = 7.2) [60]. To examine the effect of TTX on glutamate levels, we added TTX (final consternation: 1 μM, FUJIFILM Wako Pure Chemical Corporation, Osaka, Japan) or 0.9% NaCl water (saline) as a control to the MEM. We cultured the brain for 24 h at 30 ± 1.0°C in SLI-400 incubator (Tokyo Rikakikai Co., Tokyo, Japan). Further, 20 μL of the supernatant of MEM was collected without the brain. We measured the glutamate concentration in MEM using Glutamate Assay Kit-WST that can measure L-glutamate concentration ≥several μM, according to the manufacturer's instructions (Dojindo Laboratories, Kumamoto, Japan). Thirty microliters of glutamate assay working solution was added to 20 μL of MEM supernatant, and the mixture solution was incubated for 30 min at 37°C in a T-100 thermal cycler (Bio-Rad Laboratories, California, USA). Furthermore, absorbance of the solution at 450 nm was measured by DS-11 Micro-spectrophotometer (DeNovix, Wilmington, Delaware, USA). We calculated glutamate concentration using the standard curve method (glutamate concentration of some samples with negative values in the calculation was set to zero). We measured each sample in duplicate and averaged the calculated glutamate concentration.

## Electrophysiological analyses of spontaneous PI firing activity

We recorded spontaneous firing activities of PI neurons in dsRNA-injected female *R. pedestris* according to a previous report [32]. We used dsRNA-injected females 20 to 22 days after eclosion under the long-day or short-day conditions. Our previous study showed that the PI neurons did not show a drastic daily change in their spontaneous activity [32]. Thus, in this study, we recorded the neuronal activity of PI neurons from multiple samples across the daytime (ZT0–12). The female bug was anesthetized on ice and mounted in clay. Then, the whole brain was carefully desheathed and placed in a center hole (diameter: about 2 cm, depth: about 2 mm) of a handmade plastic recording chamber [54] filled with an extracellular solution (ion components: 136 mM NaCl, 4.0 mM KCl, 10 mM HEPES, 2.0 mM $CaCl_2$, 1.5 mM $MgCl_2 \cdot 6$ $H_2O$, and 10 mM glucose, pH was adjusted to approximately 7.4 with NaOH). We used a

normal intracellular pipette solution, which contained 130 mM $K^+$-gluconate, 4.0 mM NaCl, 1.0 mM $MgCl_2 \cdot 6\ H_2O$, 0.5 mM $CaCl_2$, 10 mM EGTA, and 10 mM HEPES (pH 7.2, adjusted with KOH). Additionally, we applied 20 mM neurobiotin tracer (Vector Laboratories, Burlingame, California, USA) to the intracellular solution for staining recorded neurons. We made recording pipettes from borosilicate glass capillaries (GD-1.5, Narishige, Tokyo, Japan) by a Flaming/Brown type micropipette puller (P-97, Sutter Instruments, Novato, California, USA). Tip resistance of recording pipettes in the extracellular solution was approximately 3 to 7 MΩ. Under an upright microscope (ECLIPSE FN1, Nikon, Tokyo, Japan) with an ORCA-spark digital CMOS camera (C11440-36U, Hamamatsu Photonics K.K., Shizuoka, Japan), we approached the recording pipette to the large PI cell body and formed a giga seal by negative pressure. Then, we broke the cell membrane by voltage pulse (zap) for a whole-cell patch clamp recording and recorded a spontaneous firing in a current clamp mode. The whole-cell patch recordings were stored with Axopatch 200 B, Digidata 1550 B, and pCLAMP 11.0.3 software (Molecular Devices, Sunnyvale, California, USA). A liquid junction potential was corrected off-line for all recordings. After the spontaneous firing recording was completed, neural projections of the recorded neuron were observed by labeling the neurobiotin tracer according to the previous report [32], and we confirmed that the recording cells were PI cells.

Data between 5 and 10 min after the start of recording were used for calculating an instantaneous frequency (reciprocal of an interval between spikes) and number of firings. PI neurons showed various firing patterns, and we classified firing patterns into 3 types referring to the previous study [32]. Burst: a high frequency burst firing (4 or more consecutive action potentials within 500 ms) was found. Non-burst: multiple spontaneous firing activities, but not the burst firings, were found. Silent: there were no multiple spontaneous firings in 5 min. In silent neurons, after spontaneous recordings, we confirmed an appearance of firing activities induced by positive current injection to determine whether the recording was successful. Spontaneous firing activities were analyzed by Clampfit software version 10.7 (Molecular Devices, Sunnyvale, California, USA).

## Glutamate perfusion experiments

We used intact females 20 to 22 days after eclosion under long-day conditions. The unsheathed brain was placed in a recording chamber filled with extracellular solution. Perfusion was performed using Peristaltic Pump/MINIPULS 3 (M&S Instruments, Osaka, Japan). To examine the effects of glutamate on PI spontaneous firing, we performed current clamp whole-cell patch recordings using a normal intracellular pipette solution. The previous electrophysiological study in the *Drosophila* large ventrolateral neurons showed that bath application of 1 mM glutamate had a sufficient effect on their neural activity via GluCl [61]. Also in the *Xenopus* oocytes expressing GluCl by RNA injections, dose-dependent increase in glutamate-induced currents became a plateau at about 1 mM [38]. Thus, to investigate the sufficient effect of glutamate on PI cells, we performed a single perfusion of glutamate at a high concentration of 1 mM, and 1 mM L-glutamic acid monosodium salt hydrate solution (L-Glutamate, Sigma-Aldrich, Missouri, USA) was perfused within the extracellular solution for 4 min after 8 min of baseline recording. After glutamate perfusion, we perfused the normal extracellular solution for wash-out. Instantaneous frequency and number of firing at 0.5 min to 3.5 min before the L-Glutamate perfusion were calculated as "Before," those at 0.5 min to 3.5 min after the L-Glutamate perfusion were calculated as "L-Glutamate," and those at 10 min to 13 min after the start of wash-out were calculated as "wash-out." As a negative control, we also performed a perfusion experiment of D-Glutamate (FUJIFILM Wako Pure Chemical Corporation, Osaka, Japan) by intact females 20 to 25 days after eclosion under long-day conditions. The effect of 1

mM D-Glutamate on the neural activity of PI cells was analyzed by the same experimental procedure as L-Glutamate perfusion.

To analyze glutamate-induced currents, we performed voltage-clamp whole-cell recordings. We used 2 intracellular solutions: normal intracellular pipette solution as described in a previous subsection and high Cl⁻ solution containing 130 mM KCl, 4.0 mM NaCl, 1.0 mM MgCl$_2$·6 H$_2$O, 0.5 mM CaCl$_2$, 10 mM EGTA, and 10 mM HEPES (pH 7.2, adjusted with KOH). We perfused 1 mM L-Glutamate for 1 min at each holding potential (−80 mV to −20 mV or −80 mV to 0 mV). We set the voltage-clamp recording at 1 min before the L-Glutamate perfusion as "Baseline current" (if it was overlapped with spontaneous action currents at the time point, we avoided them to calculate "Baseline current") and calculated the glutamate-induced current as follows: ([the highest/lowest current during 1 min glutamate administration] minus [Baseline current]).

We also performed voltage-clamp whole-cell recordings to analyze the effect of GluCl inhibitor picrotoxin (Sigma-Aldrich, Missouri, USA) on glutamate-induced currents. We used the normal intracellular pipette solution and set a holding potential at −20 mV. We first perfused 300 μM L-Glutamate for 1 min (1st L-Glutamate). After wash-out for more than 5 min, we next perfused 100 μM picrotoxin for 2 min, and then co-perfused 300 μM L-Glutamate and 100 μM picrotoxin for 1 min (L-Glutamate + Picrotoxin). Finally, after wash-out for more than 5 min, we perfused 300 μM L-Glutamate for 1 min (2nd L-Glutamate). Each glutamate-induced current was calculated as described above.

All analyses were performed using Clampfit software version 10.7 (Molecular Devices, Sunnyvale, California, USA).

## Blast search for target genes in *Riptortus pedestris*

We searched predicted sequences of *got*, *gs*, *glucl*, *gad*, *eaat2*, and *vglut* of *R. pedestris* from RNA sequencing data by Dr. Hiroko Udaka and Dr. Hideharu Numata (Kyoto University, Japan). For basic local alignment search tool (BLAST) analyses, we used amino acid sequences of GOT1 (NCBI/GenBank/NP_611086.1), GS2 (NCBI/GenBank/NP_996408.1), GluCl (NCBI/GenBank/ABG57261.1), EAAT2 (NCBI/GenBank/NP_477427.1), and VGLUT (NCBI/GenBank/NP_608681.2) in *D. melanogaster* and GAD (NCBI/GenBank/XP_044263918.1) in *Tribolium madensas* queries. We performed tblastn search by NCBI BLAST+ docker image version 2.11.0. By BLAST search and detected contigs of *R. pedestris* including sequences of GOT (DDBJ/GenBank/EMBL, accession number: ICRD01122828), GS (DDBJ/GenBank/EMBL, accession number: ICRD01077913), GluCl isoform1 (GluCl i1) (DDBJ/GenBank/EMBL, accession number: ICRD01079040), GluCl isoform2 (GluCl i2) (DDBJ/GenBank/EMBL, accession number: ICRD01079041), GAD (DDBJ/GenBank/EMBL, accession number: ICRD01131820), EAAT2 (DDBJ/GenBank/EMBL, accession number: ICRD01130052), VGLUT (DDBJ/GenBank/EMBL, accession number: ICRD01077818). We detected predicated open reading frame by NCBI ORF finder (https://www.ncbi.nlm.nih.gov/orffinder/). Predicated sequences and deduced amino acids were illustrated in S4, S7 and S9 Figs.

## Single-cell reverse transcription-nested PCR

We performed single-cell reverse transcription-nested PCR based on the protocol described in a previous study by our group [32]. We collected 4 PI cells from each hemisphere (8 cells per female) from 5 intact females 21 days after eclosion (labeled Female a–e). Females were maintained in the short-day conditions before eclosion and the long-day conditions after eclosion. We anesthetized females on ice and mounted in clay. The whole brain was carefully desheathed and took to the handmade chamber filled with the extracellular solution. Pipettes

for collecting cells were made from borosilicate glass capillaries (GD-1.5, Narishige, Tokyo, Japan) using a Flaming/Brown type micropipette puller (P-97, Sutter Instruments, Novato, California, USA). Under an upright microscope (ECLIPSE FN1, Nikon, Tokyo, Japan) with an ORCA-spark digital CMOS camera (C11440-36U, Hamamatsu Photonics K.K., Shizuoka, Japan), we identified target large PI cells. We isolated and sucked up a single large PI cell by the collecting pipette and placed it into a reverse transcription solution [4 μL FastGene cDNA Synthesis 5× ReadyMix (Nippon Genetics, Tokyo, Japan) + 16 μL pure water]. We performed reverse transcription and synthesized cDNA using T-100 thermal cycler (Bio-Rad Laboratories, California, USA) and TaKaRa PCR Thermal Cycler Dice (Takara Bio, Shiga, Japan). We performed nested PCR with a minor modification of our previous studies [32,54]. We prepared a PCR mix solution [1 μL template cDNA + 12.5 μL KAPATaq Extra Hot Start Ready-Mix with dye (Kapa Biosystems-Roche, Basel, Switzerland) + 0.25 μL Forward and Reverse primers (20 μM) + 11 μL pure water]. The primary and secondary PCR were composed of an initial heat denaturation at 95˚C for 3 min and 35 cycles of denaturation at 95˚C for 30 s, 48˚C for 30 s, and 72˚C for 40 s. One μL reverse transcription reaction solutions were used as template cDNA in the primary PCR. One μL primary PCR reaction solutions were used as template DNA in the secondary PCR. After the nested PCR, an electrophoresis was performed in 1.5% agarose gel using submarine electrophoresis device MARINE23ST (FUJIFILM Wako Pure Chemical Corporation, Osaka, Japan). In the electrophoresis, a 50 bp DNA ladder (product code: NE-MWD50, Nippon Genetics, Tokyo, Japan) was used to measure the length of PCR products. Then, we incubated the agarose gels in a Midori Green Advance solution (product code: NE-MG04, Nippon Genetics, Tokyo, Japan) for more than 60 min. Using Gel Documentation System AE-6932GXCF with CCD camera Controller AE-6905CF (ATTO Corporation, Tokyo, Japan), we took photographs of PCR product bands in the agarose gels. We printed the photographs by the VIDEO GRAPHIC PRINTER UP-897MD (SONY, Tokyo, Japan). Primer sets of primary and secondary nested PCR for each gene (*tubulin*, *glucl)* were listed in S2 Table.

## Statistical analysis

Kyplot 6 software (KyensLab, Tokyo, Japan) was used for statistical analyses: unpaired two-tailed *t* test, two-tailed Mann–Whitney *U* test, $\chi^2$ test, one-way analysis of variance (one-way ANOVA), Tukey–Kramer test, repeated Friedman test, Steel–Dwass test. Parametric analyses (unpaired two-tailed *t* test, one-way ANOVA, Tukey–Kramer test) were used for normally distributed data. Data distribution was analyzed by Shapiro–Wilk test (using Kyplot 6 software). Tukey-type multiple comparisons for proportions were performed using Microsoft Excel 2016 (Microsoft Corporation, Redmond, Washington, USA). $P < 0.05$ was considered to be statistically significant.

## Supporting information

**S1 Fig. Ovarian development of intact female *R. pedestris* under long-day and short-day conditions.** The rate of ovarian development was much higher under long-day conditions than under short-day conditions 20–22 days after eclosion in both daytime (A, zeitgeber time: ZT3–5) and nighttime (B, ZT16–18) dissection. $\chi^2$ test; *** $P < 0.001$. The underlying data can be found in the S1 Data datasheet of numerical values for each fig.xlsx.
(TIF)

**S2 Fig. Effects of RNAi on expression levels of *period* in the brain.** Columns with scatter plots show relative expression levels of *period* (*per*), which were normalized by expression of

*beta-tubulin* (*tubulin*), in dsRNA for *β-lactamase* (ds*bla*) and dsRNA for *per* (ds*per*)-injected females. The relative expression levels of *per* were significantly lower in the ds*per*-injected group than in the control ds*bla*-injected group. Columns with error bars show mean ± SEM. Unpaired two-tailed *t* test, *** $P < 0.001$. The underlying data can be found in the S1 Data datasheet of numerical values for each fig.xlsx.
(TIF)

**S3 Fig. Ovarian development of dsRNA-injected females under long-day and short-day conditions.** In control ds*bla*-injected females, the rate of ovarian development was much higher under long-day conditions than under short-day conditions. On the other hand, most ds*per*-injected females developed their ovaries even under short-day conditions, and there was no significant difference between long-day and short-day conditions within the ds*per* females. These dsRNA-injected females were used to measure brain extracellular glutamate concentration (Fig 1D), except for one ds*bla*-injected female under short-day conditions whose brain could not be successfully extracted. Columns with different letters show statistically significant differences (Tukey-type multiple comparisons for proportions, $P < 0.05$). The underlying data can be found in the S1 Data datasheet of numerical values for each fig.xlsx.
(TIF)

**S4 Fig. Predicted sequences for nucleotides and deduced amino acids of the open reading frame of GOT and GS in *R. pedestris*.** The symbol of # means a stop codon.
(TIF)

**S5 Fig. Effects of RNAi on expression levels of *got* and *gs* in the heads.** Columns with scatter plots show relative expression levels of *got* (A) and *gs* (B), which were normalized by expression of reference gene *tubulin*. dsRNA for *got* (ds*got*) and that for *gs* (ds*gs*) specifically decreased the expression level of each target gene under both long-day (LD) and short-day (SD) conditions. Columns with error bars show mean ± SEM. Columns with different letters show statistically significant differences (Tukey–Kramer test, $P < 0.001$). The underlying data can be found in the S1 Data datasheet of numerical values for each fig.xlsx.
(TIF)

**S6 Fig. Effects of D-Glutamate perfusion on the spontaneous firing activity of PI neurons.** (A) A representative trace showing effects of 1 mM of D-Glutamate perfusion on the spontaneous firing activity of PI neurons. (B, C) Line graphs showing the (B) instantaneous frequency and (C) number of firing events in 3 min of "Before," "D-Glutamate," and "Wash-out" within each PI cell ($n = 8$). Perfusion of D-Glutamate did not have significant effects on the spontaneous neural activity of PI neurons, whereas slightly attenuated the neural activity in some cells. Steel–Dwass test, N.S.: not significant. The underlying data can be found in the S1 Data datasheet of numerical values for each fig.xlsx.
(TIF)

**S7 Fig. Predicted sequences for nucleotides and deduced amino acids of the open reading frame of GluCl isoform1, 2 (i1, i2) in *R. pedestris*.** Red character sequences are the common sequence to the 2 isoforms. The symbol of # means a stop codon.
(TIF)

**S8 Fig. Effects of RNAi on expression levels of *glucl* in the head.** Columns with scatter plots show relative expression levels of *glucl*, which were normalized by the expression of reference gene *tubulin*. The relative expression levels of *glucl* were significantly lower in the dsRNA for *glucl* (ds*glucl*)-injected females than in the control ds*bla*-injected females under both long-day and short-day conditions. Columns with error bars show mean ± SEM. Columns with

different letters show statistically significant differences (Tukey–Kramer test, $P < 0.001$). The underlying data can be found in the S1 Data datasheet of numerical values for each fig.xlsx.
(TIF)

**S9 Fig. Predicted sequences for nucleotides and deduced amino acids of the open reading frame of GAD, EAAT2, and VGLUT in *R. pedestris*.** The symbol of # means a stop codon.
(TIF)

**S10 Fig. Brain mRNA expression of glutamate-related enzymes and transporters.** Box and scatter plots showing relative expression levels of (A) *got*, (B) *gs*, (C) *gad*, (D) *eaat2*, (E) *vglut*, which were normalized by expression of *beta-tubulin* (*tubulin*), in intact females under long days and short days. Lines at the top, middle, and bottom of the box plots indicate the upper quartile, median, and lower quartile, respectively. Upper and lower whiskers of the box plots indicate the maximum and minimum values, respectively. Two-tailed Mann–Whitney *U* test, $^* P < 0.05$, N.S.: not significant. The underlying data can be found in the S1 Data datasheet of numerical values for each fig.xlsx.
(TIF)

**S1 Table. List of *tubulin* and *glucl* expression in the large PI cells of *R. pedestris* by single cell nested PCR.**
(DOCX)

**S2 Table. Primer list for dsRNA synthesis, qPCR, single cell nested PCR.**
(DOCX)

**S1 Data. Spreadsheet containing the underlying numerical data for figures.**
(XLSX)

**S1 Raw image. Original image showing PCR product bands in an agarose gel in Fig 6C.**
(PDF)

## Acknowledgments

We thank Dr. Hiroko Udaka and Dr. Hideharu Numata (Kyoto University) for providing the RNA-sequencing data and Mr. Ryo Hashimoto (Osaka University) for establishing the BLAST analysis system.

## Author Contributions

**Conceptualization:** Masaharu Hasebe.

**Data curation:** Masaharu Hasebe.

**Formal analysis:** Masaharu Hasebe.

**Funding acquisition:** Masaharu Hasebe.

**Investigation:** Masaharu Hasebe.

**Methodology:** Masaharu Hasebe, Sakiko Shiga.

**Project administration:** Masaharu Hasebe, Sakiko Shiga.

**Supervision:** Masaharu Hasebe, Sakiko Shiga.

**Validation:** Masaharu Hasebe, Sakiko Shiga.

**Visualization:** Masaharu Hasebe.

**Writing – original draft:** Masaharu Hasebe.

**Writing – review & editing:** Sakiko Shiga.

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
