## [Editor Report · Decision Letter 0]

11 Jan 2022

Dear Dr Hasebe, 

Thank you for submitting your manuscript entitled "Circadian clock-dependent glutamate dynamics in the brain mediates photoperiodic responses in the bean bug" for consideration as a Research Article by PLOS Biology.

Your manuscript has now been evaluated by the PLOS Biology editorial staff, as well as by an academic editor with relevant expertise, and I am writing to let you know that we would like to send your submission out for external peer review.

Once your full submission is complete, your paper will undergo a series of checks in preparation for peer review. Once your manuscript has passed the checks it will be sent out for review. To provide the metadata for your submission, please Login to Editorial Manager (https://www.editorialmanager.com/pbiology) within two working days, i.e. by Jan 13 2022 11:59PM.

If your manuscript has been previously reviewed at another journal, PLOS Biology is willing to work with those reviews in order to avoid re-starting the process. Submission of the previous reviews is entirely optional and our ability to use them effectively will depend on the willingness of the previous journal to confirm the content of the reports and share the reviewer identities. Please note that we reserve the right to invite additional reviewers if we consider that additional/independent reviewers are needed, although we aim to avoid this as far as possible. In our experience, working with previous reviews does save time. 

If you would like to send previous reviewer reports to us, please email me at ggasque@plos.org to let me know, including the name of the previous journal and the manuscript ID the study was given, as well as attaching a point-by-point response to reviewers that details how you have or plan to address the reviewers' concerns. 

Given the disruptions resulting from the ongoing COVID-19 pandemic, please expect some delays in the editorial process. We apologise in advance for any inconvenience caused and will do our best to minimize impact as far as possible.

Kind regards,

Gabriel

Gabriel Gasque

Senior Editor

PLOS Biology

ggasque@plos.org

---

## [Decision Letter · Decision Letter 1]

11 Feb 2022

Dear Dr Hasebe,

Thank you for submitting your manuscript "Circadian clock-dependent glutamate dynamics in the brain mediate photoperiodic responses in the bean bug" for consideration as a Research Article at PLOS Biology. Your manuscript has been evaluated by the PLOS Biology editors, by an Academic Editor with relevant expertise, and by three independent reviewers.

In light of the reviews (below), we will not be able to accept the current version of the manuscript, but we would welcome re-submission of a much-revised version that takes into account the reviewers' comments. We cannot make any decision about publication until we have seen the revised manuscript and your response to the reviewers' comments. Your revised manuscript is also likely to be sent for further evaluation by the reviewers.

We expect to receive your revised manuscript within 3 months. 

**IMPORTANT - SUBMITTING YOUR REVISION**

1. A 'Response to Reviewers' file - this should detail your responses to the editorial requests, present a point-by-point response to all of the reviewers' comments, and indicate the changes made to the manuscript. \\

*Re-submission Checklist*

*Published Peer Review*

*PLOS Data Policy*

*Blot and Gel Data Policy*

Sincerely,

Gabriel

Gabriel Gasque

Senior Editor

PLOS Biology

ggasque@plos.org

REVIEWS:

Reviewer #1: The manuscript by Hasebe and Shiga presents the surprising finding that bean bug whole brain glutamate efflux is sensitive to photoperiod and makes the case that this per gene dependent response acts via inhibitory glutamate receptors to alter neuronal excitability and reproductive behavior. In experimental systems such as Drosophila in which identified cells can be targeted emphasis has been on neuron connectivity and selective effects of neurotransmitters and neuropeptides in defined circuits. Here glutamate and gene expression are studied and manipulated at the whole brain level, which would usually be predicted to not be very informative. Thus, data supporting a role for a global change in glutamate in circadian behaviors and neuronal activity are surprising and interesting. Unfortunately, there are some aspects of the presentation and analysis that raise serious doubts. 

First, a recurrent theme in the study is to compare data obtained from animals exposed to either "long-day" or "short-day" photoperiods. In many of the figures, the main point is that perturbing the system disrupts the differences seen in these two conditions. However, Figure 2A is presented differently than all of the other figures and the key comparison mentioned above is never made. If those differences are not significant in the control of Figure 2A, then the whole study is compromised. Furthermore, even when the presented statistical analysis shows that the difference is lost upon an RNAi treatment, the data still seem to show differences that are simply less dramatic. If paired tests were done for each manipulation, would differences between "long-day" or "short-day" photoperiods be significant? Could statistical insignificance simply reflect that irrelevant multiple comparisons were included that led p values to be inflated in a manner that does not relate to the hypothesis being tested? By that view, photoperiodic changes are affected by glutamate, but the changes are not mediated by glutamate (i.e., as stated in the title). 

Another issue is that the basis of the change in glutamate with photoperiod is unknown. Therefore, it remains possible that the whole brain data are correlative and not indicative of causation. In other words, it still remains possible that the real connection between glutamate and behavior arises at a few synapses (e.g., involving PDF neurons) and that those synapses are affected by the global perturbations used in the study. Even if the concerns presented in the previous paragraph were addressed, the data would be consistent with the authors' interpretation, but certainly the alternative has not been excluded. Given the concerns about presentation, interpretation and gaps in the understanding of the underlying biology, this manuscript does not seem currently acceptable.

Minor points:

1. 12 hour light:12 hour dark conditions are not considered short-day. In the circadian field, short-day conditions would be in the winter when most of the day is dark.

2. Line 210, suggest changing firing rate to integrated activity.

3. Line 223, suggest changing receives to binds.

4. Lines 232 and 240, suggest changing inverting potential to reversal potential.

5. Line 294, suggest changing whereas to however.

6. Line 298, suggest changing is to could be.

7. Throughout the manuscript, the text refers to brain glutamate release, but the discussion admits that changes could reflect changes in uptake. It would be better to make this admission when introducing the assay. 

Reviewer #2: In this manuscript Hasebe and Shiga examine the mechanisms underlying photoperiodic control of female reproductive diapause in the bean bug, Riptortus pedestris. The work is based on a striking induction of reproductive diapause by equinox conditions when compared to long day conditions (LD 16:8). The results are consistent with photoperiodic control of the levels of glutamate release, with higher levels of glutamate associated with equinox. The authors show that genetic manipulation of glutamate signaling disrupts induction of diapause by equinox. Interestingly, the authors show that injection RNA-interference constructs targeting the clock gene period disrupts photoperiodic differences in both glutamate levels and diapause, consistent with the central conclusions of the study that photoperiodic control of diapause is mediated by both the circadian clock and glutamate signaling. The relationships between the circadian system and photoperiodism are of great interest to the field as are mechanisms underlying such relationships. The work is therefore of significant interest. Addressing the following concerns would improve this promising work:

Major Concerns

1.) A major component of this study is the identification of glutamate as a neurochemical mediator of photoperiodism in the bean bug, but there are several aspects of the glutamate work that are cause for concern. First, the authors do not provide or cite evidence that the glutamate release assay is in fact a sensitive assay for release. Though it is certainly possible that this assay reflects relative amounts of glutamate being released by the explanted and desheathed brain, it is disconcerting that no evidence for this is present or cited, nor are alternative explanations for apparent photoperiodic differences discussed/eliminated. A critical alternative explanation, given the title and central conclusions of the study, is that there is a daily rhythm of glutamate abundance and that the phase of this rhythm is different between LD 12:12 and 16:8, thereby producing what looks like a difference in overall levels at the single time-point compared. It is well known that phases of entrainment differ between different photoperiods, but the authors have sampled brains to estimate glutamate concentrations at the same Zeitgeber Time for both conditions (ZT3-5). How can the authors be certain that the difference in glutamate concentration between short and long days is not simply a consequence of sampling out-of-phase glutamate release oscillations? It also seems possible that the assay doesn't reflect natural levels of release at all, but rather differences in the response of the brain to dissection. Might Ca2+ free media with high divalents, or some other intervention, be used to test if this assay really addresses "release?" Such conditions should prevent or drastically reduce release and result in much lower levels of GLU in the media than controls. If the authors cannot site evidence that their approach addresses release, they should provide such evidence. Showing that the other major neurotransmitters fail to show these photoperiodic differences would provide further support for the central model that glutamate release is modulated by photoperiod.

2.) A strength of this study is that the authors have directly addressed the extent to which RNAi has resulted in a reduction in the expression levels of targeted genes. Based on the data confirming knockdown, it appears that such knockdowns were partial reductions in expression levels. In the case of GLU signaling manipulations, this makes sense, a complete loss of GLU signaling would certainly be fatal, and large reductions would be expected to have large and varied behavioral effects. The authors should comment on this. But more critically, it is not clear if the period knockdown described succeeded in abolishing circadian rhythms. Has the bug's clock been stopped by period RNAi? This could be determined by locomotor rhythms under constant condtions. It's possible that the partial knockdown of period mRNA does not stop the clock, which would make the relationship between circadian timekeeping and photoperiodic diapause induction less clear for this study. There are established post-transcriptional mechanisms in insect clocks that could support rhythmicity without normal levels of period mRNA or even mRNA rhythms. Thus, the relationship between the circadian clock and mechanisms of photoperiodic diapause induction are not as clear as suggested by the title of the manuscript, which claims that photoperiodic responses are mediated by circadian clock dependent glutamate dynamics.

Other Concerns

a.) In Figure 3, the authors talk about firing patterns of neurons in the PI. The methods section says that electrophysiology experiments were done during the daytime (ZT00-12), but it is not clear if the results shown are average across multiple samples in the 12-h window or are they sampled in a different way. This needs some explanation in the text. Further both the got- and gs-RNAis seem to have the same effect on long- and short-day firing patterns of the PI neurons. Is this expected given how they are associated with reciprocal levels of glutamate release in short and long days? The authors should discuss this.

b.) In Figure 3B and 6D the relative firing proportions appear to have patterns that are different between the respective RNAis and photoperiods but one reason for not detecting statistical significance could be the use of chi-square goodness of fit tests (as inferred from the methods section). The N in these cases is too small for an accurate estimate from chi-square tests. Instead, if the authors use Fisher's exact tests, they may find statistical significance that may help their case.

c.) The authors mention in the discussion (Line 357) that previous studies have shown that ablation of PI did not disrupt photoperiodic responses in the bean bug (Ref. 33). This is surprising given the focus here on photoperiodic changes in PI neuron firing activity and their known role in oviposition (Ref. 30). The authors should put this in context and explain why the electrophysiological properties of PI neurons is of interest here if they are not necessary for the phenomenon being examined.

d.) There is no clarity on how statistics were performed in Fig. 4C-D. This is important for conclusions drawn in this section of the results. Were before, during perfusion, and wash-out values compared pairwise.? This would imply that there are three pairwise comparisons. It is not clear from the text if the p-values for each of these comparisons was 0.05 or was it adjusted for multiple comparisons. Also, it would be helpful if the line and symbols had some degree of transparency so that all the symbols can be clearly seen, because in some cases they overlap, and all 9 traces are not clearly visible.

e.) Line 324: Should be "Specific ablations targeting…"

Reviewer #3: The overall goal of this study is to investigate the molecular and neuronal mechanisms underlying circadian clock-driven neural signals in the brain of the bean bug that convey photoperiodic information. Specifically, the authors are interested in the brain dynamics of a key neurotransmitter glutamate in different photoperiods and how glutaminergic signaling could be contributing to the photoperiodic response. The bean bug Riptortus pedestris is an ideal model for this research question given its unambiguous photoperiodism in reproductive phenotypes, such as ovary development and oviposition. The authors observed that glutamate release levels in the whole brain were significantly higher under SD conditions, which normally promote reproductive diapause. Interestingly, this photoperiod-dependent change in glutamate level is abolished when period is knocked down. Furthermore, the authors observed that the knock down of got and gs (2 glutamate metabolizing enzyme genes) not only impact glutamate release level, but also abolished photoperiodic changes in reproductive phenotypes. The authors then leveraged electrophysiology to show that glutamate acts as an inhibitory signal to PI neurons via glutamate-gated chloride channel (GluCl), and RNAi knock down of GluCl, as well as got and gs can disrupt cellular photoperiodic responses of the PI neurons as well as reproductive diapause. In summary, the authors conclude that photoperiod regulates glutamate release in the brain of the bean bug, and the period gene is necessary to maintain this photoperiod response. They also show that photoperiodic-dependent change in glutamate dynamics plays an important role in photoperiodic control in reproductive physiology, and this is mediated via GluCl.

Major comments: 

1. The authors need a negative control for their electrophysiology experiments in Figure 4. In addition to adding L-glutamate, they need to add something that the neurons do not respond to in order to show that the response they observed is specific to L-Glutamate. Perhaps D-Glutamate or a different amino acid?

2. An additional negative control for the GluCl experiments would be the use of antagonists for GluCl. This would support that the inhibitory effects of glutamate is mediated by GluCl, and complement their genetic experiments. This is recommended but I do understand if the authors feel that their genetic experiment will suffice. 

3. There seems to be a disconnect in their model. Given that got and gs are not differentially expressed in LD vs SD, how is "period" changing glutamate concentration in the brain to achieve the difference they saw in LD vs SD. If got and gs are not the targets of clock control, the significance of their got and gs results is somewhat diminished. Could their enzyme activity be differentially expressed in LD vs SD, given that RNA expression is not equivalent to protein expression? Do the authors think they are normally involved in the photoperiodic response, or are they simply using them as "tools" to manipulate glutamate levels?

4. Line 357: Given that ablation of PI did not disrupt photoperiodic response in bean bug, why did the authors choose to investigate the direct effects of glutamate on PI neuronal activity? The authors noted that a good candidate brain region for glutamate-mediated control of ovarian development if the PL or CA. Why did they not conduct their electrophysiology experiments on PL or CA? 

Minor comments:

1. The authors should insert data points onto all bar graphs when possible. I understand that this is not possible when they are plotting ovary development (%), but it is certainly possible in Figure 1B-C, Figure S2, S5, and S7. 

2. Line 79: The authors should include more recent citations, e.g. https://pubmed.ncbi.nlm.nih.gov/31767753/

https://pubmed.ncbi.nlm.nih.gov/32541062/

3. Line 177: specify downregulated "only in SD".

4. Line 180: specify upregulated "only in LD". 

5. How did the authors determine the concentration of L-Glutamate to use in their experiments, e.g. did they do a dose-response experiment?

6. Additional of a model figure would be really helpful to the readers. 

7. How homogeneous is the efficiency of glutamate release (into MEM)? Could this have contributed to the variability in Fig 2A?

---

## [Decision Letter · Decision Letter 2]

17 Jun 2022

Dear Dr Hasebe,

Thank you for your patience while we considered your revised manuscript "Clock gene-dependent glutamate dynamics in the brain mediate photoperiodic responses in the bean bug" for publication as a Research Article at PLOS Biology. This revised version of your manuscript has been evaluated by the PLOS Biology editors, the Academic Editor and the original reviewers.

Based on the reviews and our editorial discussions, we are likely to accept this manuscript for publication, provided you satisfactorily address the remaining editorial points. Please also make sure to address the following data and other policy-related requests listed at the bottom of this email.

In the final round of peer review, two of the reviewers raised some concerns with the strength of the Fig 8 data examining vglut transporter mRNA expression changes with the photoperiod. Our editorial view is that this data should be taken out of the main manuscript given these concerns. We are happy to have you place these in the supplemental materials, revising the abstract and main body text and model accordingly to de-emphasize these results.

In addition, we'd ask that you consider a minor title change, along the lines of:

Clock-gene-dependent glutamate dynamics in the bean bug brain regulate photoperiodic reproduction

We expect to receive your revised manuscript within two weeks. 

*Published Peer Review History*

*Press*

Sincerely,

Kris

Kris Dickson, Ph.D. (she/her)

Neurosciences Senior Editor/Section Manager,

kdickson@plos.org,

PLOS Biology

DATA POLICY:

Fig 1 B-D; Fig 2A-D; Fig 3B-D; Fig 4C-D; Fig 5C; Fig6B; Fig 7A-F (NB – Fig 8 as well if moved to SF)

Supplemental Fig 1A-B; Fig 2; Fig 3, Fig 5A-B; Fig 6B-C; Fig 8

***Please also ensure that FIGURE LEGENDS in your manuscript include information on where the underlying data can be found, and ensure your supplemental data file/s has a legend. (NB: This step is often forgotten!)

***Please ensure that your Data Statement in the submission system accurately describes where your data can be found.

DATA NOT SHOWN?

Reviewer remarks:

Reviewer's Responses to Questions

PLOS authors have the option to publish the peer review history of their article (what does this mean?). If published, this will include your full peer review and any attached files.

Reviewer #1: No

Reviewer #2: No

Reviewer #3: No

Reviewer #1: In this revision the authors show that glutamate release from the bean bug brain explant is affected by lighting schedule. Furthermore, this release is sensitive to TTX and period gene expression. Finally, this glutamate release is implicated in behavior by knocking down glutamate metabolizing enzymes and an inhibitory ionotropic glutamate receptor, which is shown to alter electrical activity in select neurons. Whether the effects of these global perturbations reflect a widespread mechanism or an effect on a few key clock neuron synapses remains to be determined. All of the above findings are convincing, but the manuscript closes with gene expression data in Figure 8 that are overinterpreted: the authors focus on a statistically significant change in vglut mRNA, but the effect is tiny. It is known that such a small effect on changing transporter expression is minimal on transmission at a single synapse. Therefore, it is unlikely to be functionally important unless the data reflect a large change in a limited set of neurons. Obviously, the authors do not know that and furthermore have not perturbed vglut expression to assess its role in behavior. Thus, this vglut result is too preliminary and its current consideration weakens the presentation. Also, the authors may be making a mistake in their reasoning: implication of per suggests that the clock running by changes in transcription factors is needed, but it does not imply that the changes in glutamate reflect a change in gene expression. Rather, it may be that sensory inputs act independently of gene expression to interact with the gene expression-dependent clock neurons to change behavior. In this view, their search for a change in gene expression to explain the effect on glutamate release may be misguided. Overall, there are some interesting results here, particularly with respect to implicating glutamate release in rhythmic behavior. But the consideration of the data should be more cautious. Regarding the vglut data, it should be relegated to the supplemental data or not included in this manuscript because this aspect is too preliminary and not convincingly linked to the main findings. Likewise, the suggestion that it underlies the changes in glutamate should be removed form the figure and deemphasized in the text. 

Reviewer #2: In this revised manuscript, the authors have addressed my major concerns directly through new experiments and observations or by providing additional information supporting their rationale and interpretation of results. 

In summary:

In response to the concern that they had not established the extent to which glutamate collected from the culture media reflected "release," the authors provide strong evidence that it does, using TTX to inhibit AP firing and showing a significant reduction on glutamate. 

In response to the concern that differences in glutamate levels between photoperiods might simply reflect a changed phase in glutamate abundance rhythms, the authors have added data for an additional diurnal timepoint.

In response to the concern that the knockdown of Period might be incomplete and therefore not lead to a loss of circadian timekeeping, the authors state that the same experimental manipulation stops the circadian rhythm in cuticle deposition.

The authors have also provided clear responses to the minor concerns voiced in my critique of the original manuscript. 

The authors have therefore done an admirable job responding to these concerns and I have no further issues with the work, which is a very nice contribution. 

Reviewer #3: I applaud the authors for addressing my previous comments and conducting additional experiments to strengthen the manuscript. The addition of the vglut expressions data (that it changes with photoperiod based on new mRNA data) is exciting. However, in looking over the newly added model figure (which is a great addition), there is a clear disconnect; specifically the authors did not report any data linking period gene function to changes in vglut expression in long vs short day conditions. Given that the authors can target period for knock down using RNAi, I would suggest the authors examine the expression of vglut in the presence or absence of dsper in long vs short day conditions.

---

## [Editor Report · Decision Letter 3]

30 Jun 2022

Dear Dr Hasebe,

Thank you for the submission of your revised Research Article "Clock-gene-dependent glutamate dynamics in the bean bug brain regulate photoperiodic reproduction" for publication in PLOS Biology. On behalf of my colleagues and the Academic Editor, Paul Shaw, I am pleased to say that we can in principle accept your manuscript for publication, provided you address any remaining formatting and reporting issues, and one additional editorial request. The formatting issues will be detailed in an email you should receive within 2-3 business days from our colleagues in the journal operations team; no action is required from you until then. Please note that we will not be able to formally accept your manuscript and schedule it for publication until you have completed any requested changes.

Editorially, we noted that your figure legends do not yet have a statement including where the underlying data can be found. Please add in a sentence for each relevant figure saying ""The underlying data can be found in the S1 Data datasheet of numerical values for each fig.xlsx." This will need to be done for all of the relevant main and supplemental figures (All main figures 1-7; Supplemental figures 1,2,3,5,6,8 and 10). This additional information can be added during the formatting and reporting stage.

PRESS

We frequently collaborate with press offices. You can now notify your Press Office about your upcoming paper at this point, to enable them to help maximize its impact. If the press office is planning to promote your findings, we would be grateful if they could coordinate with biologypress@plos.org - they can provide more information on the Embargo date.

Sincerely, 

Kris

Kris Dickson, Ph.D. (she/her)

Neurosciences Senior Editor/Section Manager

PLOS Biology

kdickson@plos.org